# Multi-Agent Collaboration via Evolving Orchestration

**Yufan Dang**[★†]   **Chen Qian**[♣†]   **Xueheng Luo**[★]   **Jingru Fan**[★]   **Zihao Xie**[★]
**Ruijie Shi**[★]   **Weize Chen**[★]   **Cheng Yang**[♠]   **Xiaoyin Che**[◇]   **Ye Tian**[♡]
**Xuantang Xiong**[♡]   **Lei Han**[♡]   **Zhiyuan Liu**[★✉]   **Maosong Sun**[★✉]
[★]Tsinghua University   [♣]Shanghai Jiao Tong University
[♠]Beijing University of Posts and Telecommunications
[◇]Siemens   [♡]Tencent Robotics X
dangyf25@mails.tsinghua.edu.cn   qianc@sjtu.edu.cn
liuzy@tsinghua.edu.cn   sms@tsinghua.edu.cn

## Abstract

Large language models (LLMs) have achieved remarkable results across diverse downstream tasks, but their monolithic nature restricts scalability and efficiency in complex problem-solving. While recent research explores multi-agent collaboration among LLMs, most approaches rely on static organizational structures that struggle to adapt as task complexity and agent numbers grow, resulting in coordination overhead and inefficiencies. To this end, we propose a puppeteer-style paradigm for LLM-based multi-agent collaboration, where a centralized orchestrator ("puppeteer") dynamically directs agents ("puppets") in response to evolving task states. This orchestrator is trained via reinforcement learning to adaptively sequence and prioritize agents, enabling flexible and evolvable collective reasoning. Experiments on closed- and open-domain scenarios show that this method achieves superior performance with reduced computational costs. Analyses further reveal that the key improvements consistently stem from the emergence of more compact, cyclic reasoning structures under the orchestrator's evolution. Our code is available at https://github.com/OpenBMB/ChatDev/tree/puppeteer.

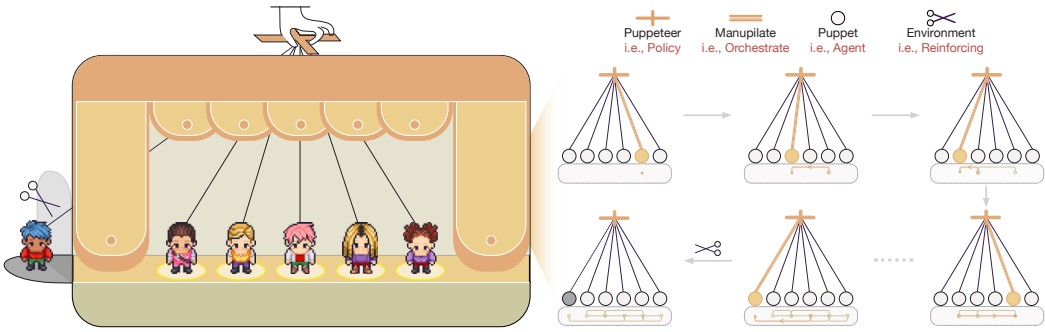

Figure 1: Overview of the proposed multi-agent collaboration framework. A central policy ("puppeteer") dynamically orchestrates which agent ("puppet") should reason at each step based on the evolving state of the task. As the task progresses, the orchestrator adaptively promotes more effective agents while removing those that are less useful, analogous to a puppeteer learning to skillfully pull or cut strings to direct a performance.

---

[†]Equal Contribution.
[✉]Corresponding Author.

39th Conference on Neural Information Processing Systems (NeurIPS 2025).

# 1 Introduction

Large language models (LLMs) [69, 48, 28] have achieved remarkable advances across diverse natural language processing tasks, demonstrating strong capabilities in planning [62, 57, 46], reasoning [70, 53, 33], and decision-making [31, 64, 1]. As the ambition to tackle ever more complex, multi-faceted problems—such as tool-augmented inference [47, 51, 86, 55] and collaborative problem-solving [42, 29, 12] in open environments—continues to grow, the limitations of monolithic LLMs are becoming increasingly apparent [71, 8, 9].

To address these challenges, recent research has drawn inspiration from human teamwork, exploring multi-agent systems (MAS) comprising diverse LLM-based agents with specialized skills [45, 52], personalized reasoning patterns [70, 76, 43], and external tool integrations [85, 47]. However, many existing approaches rely on predefined or statically generated agent topologies [61, 44] that lack flexibility and scalability. This rigidity often leads to increased coordination overhead, degraded system performance, and inefficiencies as the variety of tasks [81, 23] and the number of agents scale [45]. Especially in large-scale scenarios, the absence of principled and dynamic coordination can further result in redundant computation [77], ineffective communication [13], and diminished collective problem-solving effectiveness [6].[1]

*Can dynamic[2] orchestration simultaneously maximize collaborative effectiveness and computational efficiency?* Answering this question is crucial for building scalable, robust, and practical collective intelligence suitable for complex real-world settings. To this end, we propose a novel paradigm for constructing flexible and continually evolving multi-agent system. Drawing inspiration from puppet shows—where a central puppeteer skillfully directs multiple puppets behind the scenes—we reconceptualize multi-agent collaboration as a reasoning process orchestrated by a centralized puppeteer who dynamically selects and sequences agent activations based on evolving task states, implicitly coordinating collaboration within the group. As tasks progress, the orchestrator learns to prioritize effective agents and suppress less efficient ones, continually driving the system toward higher overall performance and efficiency.

Specifically, our framework advances multi-agent reasoning by introducing two key innovations: (i) *Dynamic Orchestration*: Moving beyond static collaboration patterns, we employ a dynamic orchestrator that routing agents at each step based on the current contexts. This process is formulated as a sequential decision problem, effectively yielding an implicit inference graph and supporting flexible, scalable agent coordination. (ii) *Adaptive Evolution*: To maximize efficiency and minimize redundancy, we employ reinforcement learning to continuously update the orchestrator's policy by leveraging feedback from completed tasks. Over time, the orchestrator learns to emphasize strong agent trajectories and prune less effective ones, enabling the system to evolve toward greater efficiency progressively.

Empirical results on both closed- and open-domain scenarios demonstrate that our approach consistently yields more effective solutions while requiring less computational overhead. Analyses further reveal that, although the evolved topologies are not fixed across different tasks, the key improvements consistently stem from the emergence of more compact, cyclic reasoning structures.

# 2 Method

We propose a unified framework for multi-agent reasoning that organizes diverse LLM-based agents via orchestrating their collaboration dynamically using a centralized policy, and continually optimizes their collaboration process through reinforcement learning.

A LLM-based agent can be abstracted in its minimal form as $a = (m, r, t)$, where $m$ denotes the foundation model, $r$ represents the reasoning pattern or prompting strategy, and $t$ is the set of available external tools. The agent space $\mathcal{A}$ enumerates all possible agents formed by combinations of these components, i.e., $\mathcal{A} = \{(m, r, t)\}$, encompassing both intrinsic and tool-augmented reasoning. Each agent thus represents an atomic reasoning behavior participating in task solving.

---

[1]For example, mesh-structured multi-agent systems with 50 nodes can require up to 10 hours to develop software comprising only a few hundred lines of code.

[2]This paper refers to the dynamic organizational structure during real-time reasoning.

For multi-agent collaboration, following [45], a MAS is naturally formalized as a directed graph $\mathcal{G} = (\mathcal{V}, \mathcal{E})$, where each node $v \in \mathcal{V}$ corresponds to an agent $a \in \mathcal{A}$, and each directed edge $(v_i, v_j) \in \mathcal{E}$ encodes a dependency or information flow, conveying intermediate context from agent $a_i$ to agent $a_j$. Typically, this graph presents a single-source, single-sink configuration: the source node represents the input task, while the sink node yields the unified task output (i.e., artifact). This formalism underlies a unified and temporal modeling framework for LLM-based reasoning systems and is analogous to a "graph-of-thought" [2] that captures the deep thinking process.

## 2.1 Dynamic Orchestration

**Centralized Puppeteer** A challenge in multi-agent reasoning is achieving efficient orchestration as task complexity and agent diversity increase. In previous approaches, each agent autonomously selects collaborators, but this incurs coordination overhead and poor scalability, particularly as agents increase or change [87, 45]. Instead, inspired by centralized coordination (*e.g.,* a puppeteer managing multiple puppets), we model the system as driven by a centralized orchestrator. This orchestrator dynamically selects which agents to activate in each step, based on the dynamic task state, and delegates reasoning to the selected agents. Such centralization decouples agent selection from internal agent behaviors, greatly enhancing adaptability and scalability without extensive parameter retraining.

**Serialized Orchestration** Another challenge stems from the combinatorially large space of possible collaboration topologies among agents. Exhaustive search is infeasible, thus prior work focuses only on canonical graphs (*e.g.,* chains, trees, graphs) [76, 70, 2]. Instead, we propose to serialize the collaborative reasoning process: rather than searching the entire topological space, the orchestrator "unfolds" the graph into a reasoning sequence guided by a topological traversal strategy. By maintaining a topological ordering, the reasoning steps follow the partial order implied by the problem dependencies. It is important to note that although the orchestration of agents appears to be serialized and "unfolded", when this episode is restored through folding, it can be reconstructed as a directed graph (with agents as nodes and orchestration partial order relations as edges).

Building on the two concepts above, we formalize the multi-agent collaboration as a sequential decision process governed by a centralized policy $\pi$. At each time step $t$, the orchestrator selects a single agent $a_t \in \mathcal{A}$ to activate, conditioned on the current global system state $S_t$ and the task specification $\tau$. The global state $S_t$ consists of all relevant agent states and aggregated contextual information up to step $t$:

$$a_t \sim \pi(S_t, \tau) = \mathbb{P}(a \mid S_t, \tau) \tag{1}$$

where the policy $\pi$ is a function mapping the observable context—such as the current state and task description—to a distribution over candidate agents [73, 74] (*e.g.,* via neural scoring, embedding-based models, or Bradley-Terry style approaches [68]).

Upon activation, agent $a_t$ receives its state $s_t(a_t)$ (extracted from $S_t$) and generates its output by a generative reasoning mapping $f_{a_t}$, after which the system state is updated ($\Phi$) as:

$$o_t = f_{a_t}(s_t(a_t), S_t), \quad S_{t+1} = \Phi(S_t, o_t) \tag{2}$$

The process continues iteratively: at each step, the orchestrator observes the updated system state $S_{t+1}$ and selects the next agent $a_{t+1}$ to activate, again using the policy $\pi$ conditioned only on $S_{t+1}$ and $\tau$. This sequential agent selection process explicitly satisfies the Markov property [17]:

$$\mathbb{P}(a_{t+1} \mid S_0, \ldots, S_{t+1}, \tau) = \mathbb{P}(a_{t+1} \mid S_{t+1}, \tau) \tag{3}$$

The process terminates when a stopping criterion is met (*e.g.,* when the selected agent is a designated terminator or when the task-solving resource is exhausted). At that point, a final aggregation function $F_{\text{agg}}$ combines all agent outputs across timesteps to yield the overall solution:

$$o^* = F_{\text{agg}}(\{o_0, o_1, \ldots, o_T\}) = \Phi(S_T, o_T) \tag{4}$$

where $T$ denotes the total number of reasoning steps.

## 2.2 Adaptive Evolution

While dynamic orchestration enables flexible agents' long-chain reasoning, naive implementations may invoke redundant or unhelpful agents, resulting in unacceptable inefficiency. To address this,

thanks to the Markov property, a learnable policy is considered for continuously learning to make agent selection decisions adaptively. After each reasoning episode, the system receives feedback jointly evaluating solution quality and resource consumption, enabling the policy to learn which agent is most valuable based on real-time task states.

Practically, this facilitates dynamic pruning of agents: the orchestration process adapts to increasingly favor compact reasoning chains by reducing reliance on agents that offer little incremental gain or incur excessive cost. Over time, the orchestrator policy evolves to organize more effective agent sequences, balancing expressive collaboration with computational efficiency. Thus, the evolvable puppeteer not only orchestrates agent collaboration, but also distills the reasoning process to its most essential components, enabling robust and scalable performance.

**Policy Optimization** To systematically optimize both the effectiveness and efficiency of collaboration, we employ REINFORCE [60], a reinforcement learning (RL) technique, as our underlying optimization framework [40, 31, 73, 54]. By doing so, the orchestration policy learns from previous executions, adaptively refining agent selection and pruning strategies to achieve more robust and cost-efficient multi-agent reasoning.

Concretely, the optimization objective is to maximize the expected return over complete reasoning trajectories, where the return reflects both overall effectiveness and inference efficiency:

$$J(\theta) = \mathbb{E}_{\pi_\theta}\big[R(\tau)\big], \quad \nabla_\theta J(\theta) \approx \frac{1}{N} \sum_{n=1}^{N} \left( \sum_{t=1}^{T} \nabla_\theta \log \pi_\theta(a_t|S_t) \right) \cdot R(\tau) \tag{5}$$

with $R(\tau)$ denoting the total reward accrued for trajectory $\tau = \{S_0, a_0, o_0, S_1, \ldots, S_T, a_T, o_T\}$, and $o_t$ the output generated by agent $a_t$ at state $S_t$, $N$ is the sample size, and $T$ is the total number of steps in one trajectory. The orchestrator's parameters $\theta$ are updated iteratively via gradient ascent: $\theta \leftarrow \theta + \alpha \nabla_\theta J(\theta)$, with learning rate $\alpha$. Through such RL-driven optimization, the orchestrator leverages accumulated cross-task experience to refine agent selection, dynamically suppressing unhelpful or costly agents and converging toward more compact, high-performing collaboration structures.

**Reward Design** To effectively guide the orchestrator's optimization, we design a reward function that jointly accounts for both solution quality and computational efficiency. Upon completion of each task trajectory, a terminal reward $r$ is assigned: for tasks with ground-truth answers, $r \in \{0, 1\}$ indicates correctness; for open-ended tasks, $r \in [0, 1]$ quantifies answer quality. The overall reward is defined recursively over time steps [59]. At the terminal state $(t = T)$, the cumulative reward incorporates both solution quality and total computational cost:

$$R_t = \begin{cases} r - \lambda \cdot C_T, & \text{if } t = T \\ \gamma \cdot R_{t+1} - \lambda \cdot C_t, & \text{if } t < T \end{cases}, \quad C_t = F \cdot \log\left(1 + \frac{t}{\varphi}\right) \tag{6}$$

where $\lambda$ controls the trade-off between accuracy and efficiency, $\gamma \in (0, 1]$ is the discount factor. To encourage economical reasoning, we penalize excessive computational expenditure. Specifically, for each reasoning step $t$, we define a step-wise cost $C_t$ based on FLOPs or token-level metrics [50], denoted by $F$, and a step factor normalized by the maximum step budget $\varphi$, i.e., $t/\varphi$. This composite reward formulation incentivizes the orchestrator to achieve correct and high-quality solutions while minimizing unnecessary computation, ultimately driving the MAS to discover reasoning structures that are both effective and cost-efficient.

## 3 Experiments

**Datasets and Metrics** To thoroughly assess our framework, we use a range of publicly available and logically demanding datasets covering both closed- and open-domain reasoning tasks:

- **Closed-domain Tasks**: These tasks require precise, objective reasoning and unambiguous answers, making them well-suited for evaluating core inference accuracy and mathematical rigor. *GSM-Hard*[18] features arithmetic problems involving unusually large numbers and complex multi-step calculations, challenging models' advanced mathematical reasoning and error-free execution. *MMLU-Pro*[67] is a comprehensive benchmark spanning diverse subjects and difficulty levels,

using multiple-choice questions to assess both factual knowledge and logical deduction. Both benchmarks are designed to assess the model's ability in mathematical and commonsense reasoning and inference, with *accuracy* as the evaluation metric.

- **Open-domain Tasks**: These tasks are inherently creative and open-ended, requiring multi-dimensional qualitative evaluation. They rigorously assess agents' ability to integrate information, understand context, and generate novel solutions. *SRDD*[44] consists of real-world textual software requirements and tasks agents with building the corresponding software, demanding proficiency in requirement comprehension, system/design reasoning, code generation, and testing. Its official evaluation metric combines completeness, executability, and consistency [44], reflecting the practical demands of real-world software development workflows. *CommonGen-Hard*[39] challenges agents to generate coherent sentences that connect seemingly unrelated concepts, highlighting their abilities in commonsense reasoning, contextual understanding, and creative expression. Evaluation is based on an aggregate metric that incorporates grammar, relevance, logical consistency[32], as well as concept coverage [39], providing a nuanced assessment of generative quality.

**Baselines** To mitigate performance interference—where strong models may overshadow the contributions of weaker ones—and to evaluate our method's adaptability across agents with varying capacities, we partition the agent pool based on the parameter scale of the underlying models. Specifically, we define a **Mimas** subspace (smaller models: *Qwen-2.5-7B*, *Qwen-2.5-14B*, *LLaMA-3.1-8B*, *LLaMA-3.2-3B*, *Mistral-7B*, *Mistral-Nemo-12B*) and a **Titan** subspace (larger models: *GPT-4-Turbo*, *GPT-4o-Mini*, *Gemini-1.5-Pro*, *Gemini-1.5-Flash*, *Claude-3-Sonnet*, *Claude-3-Haiku*, *Qwen-2.5-72B*, *LLaMA-3.1-405B*), covering both closed- and open-source families. All experiments are performed under both the Titan and Mimas subspace settings. To ensure the rigor and credibility of our experimental study, we select a suite of representative and recent baselines that comprehensively span the spectrum from straightforward LLM generation to advanced agentic paradigms:

- **Pure Models**: This category evaluates foundation models *in the absence of explicit agent structuring or workflow orchestration*, serving as a baseline for generative inference performance. For Mimas, competitive open-source models include *LLaMA-3.1-8B*, *Mistral-Nemo-12B*, and *Qwen-2.5-14B*.[3] For Titan, options such as *LLaMA-3.1-405B*, *GPT-4o-mini*, and *GPT-4-turbo* are considered.
- **Single-Agent Methods**: This category explores paradigms where reasoning is performed by a single agent using a specific reasoning pattern or workflow. *Self-refine*[39] exemplifies iterative, self-corrective reasoning within a feedback loop, whereas *AFlow*[81] enhances agent reasoning efficiency by utilizing Monte Carlo Tree Search to optimize code-represented agent workflows.
- **Multi-Agent Methods**: We benchmark against the latest multi-agent reasoning systems, showcasing state-of-the-art capabilities in leveraging agent heterogeneity and dynamic collaboration. *MacNet* [45] orchestrates agents within topologically static directed acyclic graphs, facilitating collective intelligence driven by one foundation model to enhance reasoning performance. *EvoAgent* [78] adopts evolutionary algorithms to automatically generate and optimize diverse multi-agent systems, adaptively improving collaboration and performance without manual intervention.

**Implementation Details** Different agents are equipped with distinct reasoning patterns—such as *task decomposition*, *reflection*, *refinement*, *critique*, *modification*, *summarization*, and *termination*—enabling flexible problem-solving [43, 21, 82, 39]. External tools like *WebViewer*, *WikiSearch*, *BingSearch*, *arXivSearch*, *Code Interpreter*, and *File Reader* are also integrated [47]. Dynamic collaboration uses majority voting for output aggregation. The policy is initialized with a variant of Llama-3.1[4], using default settings: episode length to 4, parallel exploration up to 3, $\lambda = 0.1$, and $\gamma = 0.99$. All baselines are rerun under identical settings.

## 3.1 Does Our Method Lead To Elevated Performance?

Many prior studies on multi-agent systems have employed the same base model to drive agent behavior [44, 36, 87]. To enable a more meaningful comparison and account for model heterogeneity, our method, referred to as Puppeteer, introduces two distinct configurations within each agent

---

[3]Other smaller-scale models in the same series (*e.g., Qwen-2.0-7B*) have also been experimentally validated, and their performance is generally weaker than that of the larger-scale model within the same series.

[4]https://huggingface.co/nvidia/Llama-3.1-Nemotron-70B-Reward-HF

subspace: the Mono setting, wherein all agents are driven by the same model, and we use *LLaMA-3.1-405B* for Titan subspace and *LLaMA-3.1-8B* for Mimas subspace, and the default setting, wherein agents are driven by a diverse set of models. As Puppeteer undergoes online reinforcement learning, we categorize its learning process into two distinct phases: an initial phase characterized by unrefined behavior, and an evolved phase marked by reinforced performance.

Table 1: Performance comparison of different methods across various datasets in Titan and Mimas subspaces respectively. For each dataset, the highest scores are highlighted in bold and the second-highest are underlined.

| Mimas | GSM-Hard | | MMLU-Pro | | SRDD | | CommonGen-Hard | | AVG. | |
|---|---|---|---|---|---|---|---|---|---|---|
| Llama-3.1-8B | $0.0000^\dagger$ | | 0.5250 | | $0.4615^\dagger$ | | $0.6992^\dagger$ | | $0.4214^\dagger$ | |
| Mistral-Nemo-12B | $0.0350^\dagger$ | | $0.4500^\dagger$ | | $0.2097^\dagger$ | | $0.7146^\dagger$ | | $0.3523^\dagger$ | |
| Qwen-2.5-14B | $0.0450^\dagger$ | | $0.3800^\dagger$ | | $0.5891^\dagger$ | | $0.5747^\dagger$ | | $0.3972^\dagger$ | |
| Self-Refine$_{Llama-3.1-8B}$ | 0.4750 | | $0.2600^\dagger$ | | $0.5412^\dagger$ | | $0.6018^\dagger$ | | $0.4695^\dagger$ | |
| AFlow$_{Llama-3.1-8B}$ | $0.2900^\dagger$ | | 0.5000 | | $0.6362^\dagger$ | | 0.7194 | | $0.5364^\dagger$ | |
| MacNet$_{Llama-3.1-8B}$ | $0.0000^\dagger$ | | $0.2000^\dagger$ | | $0.2017^\dagger$ | | **0.7434** | | $0.2862^\dagger$ | |
| EvoAgent$_{Llama-3.1-8B}$ | $0.1250^\dagger$ | | 0.5000 | | $0.2510^\dagger$ | | 0.7167 | | $0.3981^\dagger$ | |
| | Initialized | Evolved | Initialized | Evolved | Initialized | Evolved | Initialized | Evolved | Initialized | Evolved |
| Puppeteer-Mono$_{Llama-3.1-8B}$ | 0.2467 | 0.4800 | 0.4500 | 0.5200 | 0.6983 | **0.7249** | 0.6323 | 0.7341 | 0.5068 | 0.6147 |
| Puppeteer | **0.5600** | 0.5400 | 0.5700 | **0.6300** | 0.6653 | 0.6266 | 0.7139 | 0.7333 | 0.6273 | **0.6324** |

| Titan | GSM-Hard | | MMLU-Pro | | SRDD | | CommonGen-Hard | | AVG. | |
|---|---|---|---|---|---|---|---|---|---|---|
| Llama-3.1-405B | $0.1350^\dagger$ | | 0.7600 | | $0.6061^\dagger$ | | $0.8116^\dagger$ | | $0.5781^\dagger$ | |
| GPT-4o-Mini | $0.1050^\dagger$ | | $0.5950^\dagger$ | | $0.6822^\dagger$ | | $0.6691^\dagger$ | | $0.5128^\dagger$ | |
| GPT-4-Turbo | $0.2750^\dagger$ | | $0.6800^\dagger$ | | $0.6244^\dagger$ | | $0.7632^\dagger$ | | $0.5856^\dagger$ | |
| Self-Refine$_{Llama-3.1-405B}$ | 0.5250 | | $0.6000^\dagger$ | | $0.6345^\dagger$ | | $0.7003^\dagger$ | | $0.6157^\dagger$ | |
| AFlow$_{Llama-3.1-405B}$ | $0.5400^\dagger$ | | 0.7500 | | $0.6478^\dagger$ | | 0.8218 | | $0.6899^\dagger$ | |
| MacNet$_{Llama-3.1-405B}$ | $0.2905^\dagger$ | | $0.4800^\dagger$ | | $0.4228^\dagger$ | | **0.8817**$^\dagger$ | | $0.5187^\dagger$ | |
| EvoAgent$_{Llama-3.1-405B}$ | $0.4250^\dagger$ | | $0.5400^\dagger$ | | $0.1730^\dagger$ | | 0.8599 | | $0.4994^\dagger$ | |
| | Initialized | Evolved | Initialized | Evolved | Initialized | Evolved | Initialized | Evolved | Initialized | Evolved |
| Puppeteer-Mono$_{Llama-3.1-405B}$ | 0.5400 | 0.6100 | 0.6910 | 0.7600 | 0.6264 | **0.7697** | 0.8111 | 0.8417 | 0.6671 | 0.7453 |
| Puppeteer | 0.6600 | **0.7000** | 0.7400 | **0.8300** | 0.6191 | 0.7637 | 0.7381 | 0.7987 | 0.6893 | **0.7731** |

As detailed in Table 1, Puppeteer consistently achieves superior average performance during the evolved phase across all evaluated tasks, irrespective of domain type or model space size. Similarly, Puppeteer-Mono demonstrates robust performance across both large- and small-scale models. These results collectively underscore the exceptional capability of our centralized orchestrator in coordinating both heterogeneous and single-model-driven agents to form highly effective MAS, highlighting its robustness in managing diverse agent configurations.

Compared to various agent workflows and multi-agent baselines using the same base model, Puppeteer-Mono consistently outperforms competing methods across nearly all evaluated tasks. This result highlights the efficacy of our centralized orchestrator in coordinating single-model-driven agents with optimized reasoning strategies and strategic tool utilization, surpassing alternative frameworks. It underscores that superior performance stems from sophisticated organizational design and a context-aware, dynamically constructed multi-agent topology. Notably, despite Puppeteer-Mono leveraging near-optimal models within its respective subspaces, Puppeteer consistently achieves superior performance, likely benefiting from complementary interactions among heterogeneous agents driven by diverse models. Additionally, the expanded space in Puppeteer enables broader exploration of the solution landscape, thereby enhancing optimization opportunities.

To illustrate Puppeteer's capability in organizing effective MAS, we compare performance between the initial and evolved phases. The results show that continued optimization yields substantial gains—for example, Puppeteer in the Titan subspace improves from 0.6893 to 0.7731 on average, with a similar trend observed in the Mimas subspace. These findings highlight the critical role of continual optimization in enhancing coordination and task specialization, and further suggest promising directions for advancing beyond traditional, static agent paradigms toward more adaptive and collaborative agent systems.

## 3.2 Does Performance Gain Come at the Expense of Efficiency?

Recent research in non-learnable multi-agent systems has highlighted a trade-off: performance gains achieved through agent collaboration are often accompanied by increased overall token consump-

tion [77, 45]. To investigate whether our approach exhibits a similar pattern, we visualize the average token consumption and the number of orchestrated agents throughout the training process.

As shown in Figure 2, the token metric consistently decreases over the course of learning across almost all settings. This demonstrates that our system's performance improvements do not come at the cost of increased computational overhead; on the contrary, our approach achieves simultaneous advances in both effectiveness and efficiency. This result is primarily attributed to our reward design, which balances accuracy and efficiency via a tunable weighting factor $\lambda$, enabling adaptable trade-offs tailored to different application needs, with higher values of $\lambda$ indicating a greater emphasis on minimizing cost, as shown in the Figure 3. Specifically, the reward is designed to encourage the orchestrator to: (i) prioritize agents that complete tasks with reduced token usage while preserving performance, and (ii) terminate reasoning early by invoking the Terminator agent, thereby fostering efficiency through the exclusion of redundant or low-contributing agents. This

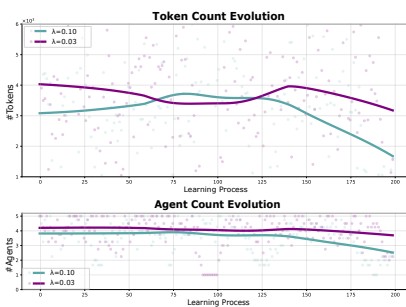

Figure 3: Token consumption and agent count orchestrated in the Titan subspace for Puppeteer-Mono on MMLU-Pro.

mechanism enables the orchestrator to optimize both overall performance and resource consumption. In extreme cases, if the efficiency factor is entirely omitted from the reward design, the system naturally degenerates into a traditional large-scale collaborative framework, albeit with potentially further improvements in performance.

More specifically, in the Titan setting, the number of active agents notably decreases over the course of learning, suggesting that the orchestrator progressively learns to terminate the reasoning process earlier for more efficient problem-solving. In contrast, in the Mimas setting (see Appendix for results), the number of orchestrated agents remains relatively stable, indicating the orchestrator's caution in prematurely halting the reasoning process due to the comparatively limited capabilities of the agents. In this case, reductions in token consumption are primarily achieved through the preferential selection of lower-cost agents rather than shorter reasoning chains. This contrast between Titan and Mimas arises from fundamental differences in agent capacity: Titan agents can solve tasks more efficiently, enabling earlier stopping without quality loss, whereas Mimas agents often require longer, more elaborate reasoning processes to ensure reliable completion.

### 3.3 How Does Organizational Topology Evolve During Reinforcement?

To elucidate the emergent organization of multi-agent collaboration, we systematically examine the evolution of agent interaction topologies throughout the learning process. Multi-agent reasoning can be abstractly modeled as dynamic orchestration governed by our centralized orchestrator, yet empirical evidence reveals an untrained "initialized" MAS system often results in a highly sophisticated and adaptive organizational structure. Instead of relying on a static structure, our Puppeteer dynamically constructs intricate topological motifs—such as trees, graphs, and cycles—by selecting the next agent to activate at each reasoning step based on the current reasoning state. Thus, the topology emerges incrementally during reasoning, embodying a flexible, context-aware organizational paradigm.

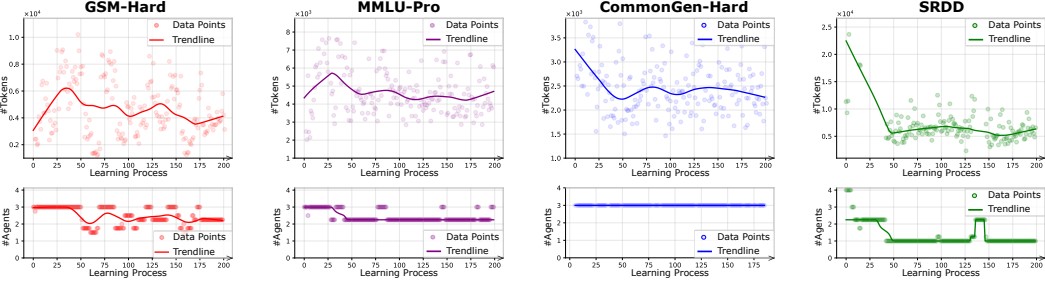

Figure 2: Evolution of token consumption and the number of orchestrated agents per task along the training process. Trends are fitted using LOWESS (Locally Weighted Scatterplot Smoothing) [14].

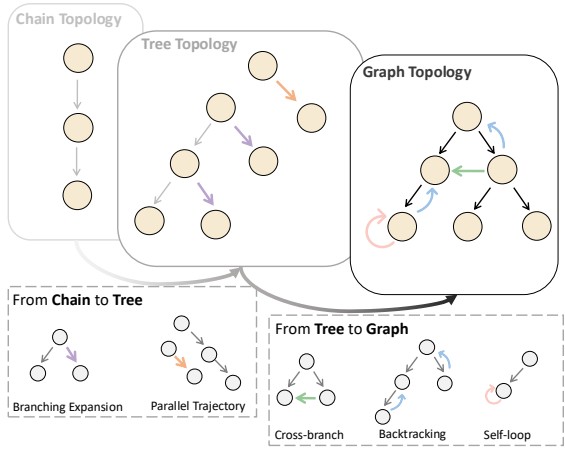

Figure 4: The organizational topology evolves as a general directed graph, shaped by the unconstrained orchestration.

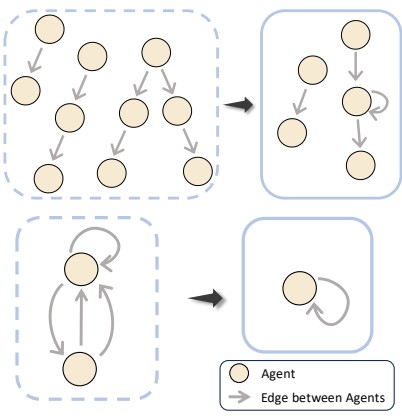

Figure 5: Examples of topology evolution.

Although the simplest form of multi-agent collaboration is often represented as a chain structure [44, 70], the Puppeteer's dynamic orchestration naturally fosters tree-structured interactions by enabling the selection of one or multiple agents at each step. This mechanism supports branching behavior and parallel pathways, thereby enhancing scalability as the number of agents grows. However, despite an initial resemblance to tree-like expansion [76] driven by branching, the resulting topologies are inherently graph-structured [45, 87, 2]. This property arises from the flexible orchestration, which permits repeated agent activations, leading to cycles and feedback loops. Moreover, cross-branch connections emerge organically, underscoring the system's capacity to generate rich, expressive, and adaptive interaction patterns. Representative examples of these emergent motifs, including cycles, backtracking, and cross-branch links, are illustrated in Figure 4.

As the puppeteer evolves over time, its orchestrating behavior changes accordingly, leading to distinct behaviors in the resulting MAS. Here, we present a specific sample, selected from both the initial and evolved phases, to demonstrate the emerging optimization effects. As shown in Figure 5, the initial phase features multiple disjoint chains, reflecting exploratory organization; after evolution, paths become fewer and cycles appear, indicating more stable and coordinated interactions. Additionally, the initial phase features two-agent communication with higher overhead; as evolution progresses, the structure condenses to a single agent handling continuous reasoning, reflecting more efficient coordination and decision-making.

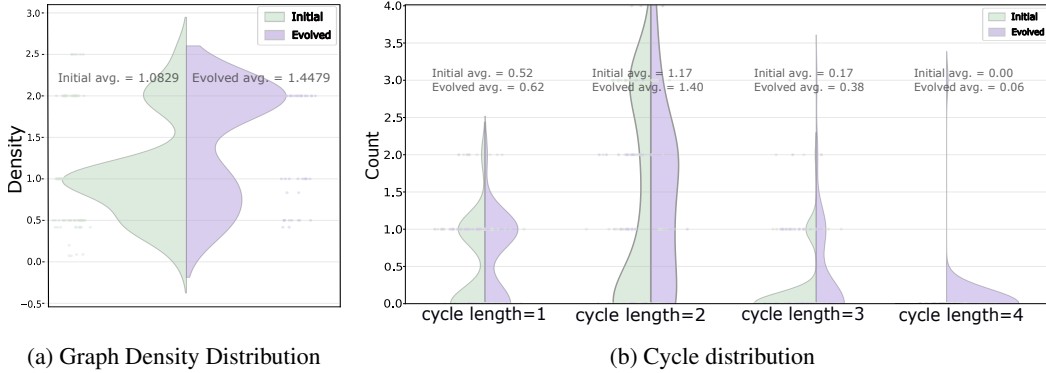

(a) Graph Density Distribution

(b) Cycle distribution

Figure 6: The compaction and cyclicality dynamics in the evolution of multi-agent organizational structures.

Empirical observation reveals that Puppeteer's dynamic orchestration—which fosters graph-structured topologies with diverse inter-agent connections—gives rise to two key structural phenomena: *compaction* and *cyclicality*. The evolving interplay between densely clustered agents and frequent communication cycles marks a significant transformation in multi-agent systems, shifting from loosely organized, exploratory interactions to tightly coordinated, specialized collectives.

- *Compaction*. We observe a marked trend toward *structural compaction* over the course of learning (Figure 6a). As optimization proceeds, graph density—quantifying the degree of inter-agent connectivity—steadily increases, with organizational structure evolving toward highly interactive and tightly coupled subnetworks. Communication becomes progressively concentrated among a subset of recurrently active 'hub' agents, forming dense subgraphs characterized by frequent and focused information exchange. This phenomenon is particularly pronounced in the Titan (large-model) subspace, where the orchestrator preferentially routes decisions through a small cohort of strong agents, thereby intensifying iterative reasoning and collaborative consensus formation. Ultimately, the system transitions from diffuse, exploratory interaction patterns to highly synergistic and focused multi-agent reasoning.

- *Cyclicality*. In conjunction with compaction, we document a significant rise in *cycle formation* as learning progresses (Figure 6b). Cyclic topologies—in which agents repeatedly revisit previous collaborators via closed-loop routes—facilitate the re-circulation of intermediate results, mutual verification, and continual refinement. Unlike strictly hierarchical or acyclic networks, this cyclical structure supports recursive critique and sustained internal debate, much like what is seen in reflexive multi-agent paradigms (*e.g.,* Reflexion [54]). As cycles become more prevalent, the system exhibits deeper internal feedback, more efficient reuse of information, and increased resilience—hallmarks of mature, self-referential collaborative reasoning.

### 3.4 How to Further Use Hyperparameters to Control Performance and Efficiency?

While reward shaping provides a direct mechanism for efficiency control, the collaboration structure itself is also crucial. As the orchestrator organizes agent collaboration, upper-bound constraints on topology—specifically, chain depth and exploration width—are essential to prevent unbounded scaling and inefficiency. Depth refers to the length of orchestrated agent chains, while width captures the number of parallel explorations. As shown in Figure 7, there is a clear non-monotonic relationship: the default setting achieves the best trade-off for our purposes, whereas increasing depth or width leads to redundancy, higher computational costs, and possible performance degradation. In general, enhancing accuracy tends to increase token consumption, and vice versa, suggesting that carefully balancing depth and width can help maintain both efficiency and effectiveness.

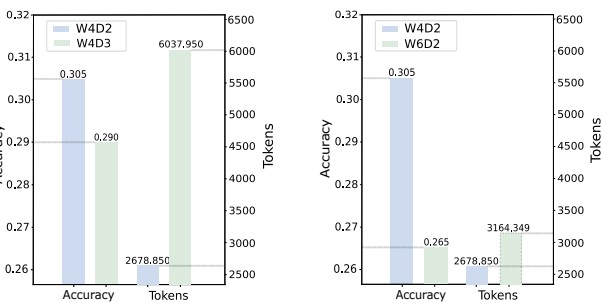

Figure 7: Impact of topology constraints on token consumption and accuracy (W$x$D$y$ denotes exploration with width $x$ and depth $y$; W4D2 is the default).

## 4   Related Work

The rapid advancement of LLMs [69, 3, 48, 28, 31, 4] has spurred the development of autonomous LLM-based agents [71, 49, 42, 29, 72, 58], which exhibit strong capabilities in planning [66, 21, 24], memory [42, 25, 41], and tool usage [51, 47, 5, 75]. These agents demonstrate growing proficiency in addressing complex tasks [19, 80, 44, 20, 62], adapting to dynamic real-world environments [83, 38, 84], and exhibiting human-like behaviors such as collaboration and decision-making [10, 42, 52]. Given that a single LLM-based agent may struggle to handle the diverse and complex range of real-world tasks [16, 29, 64], recent research has increasingly focused on constructing LLM-based multi-agent systems [11, 44, 22, 61, 12] for software development[44, 22], social simulation[42, 25], medical treatment[61, 30], scientific discovery [79].

Early MAS designs relied on fixed, handcrafted structures, *e.g.,* mirroring software engineering paradigms like waterfall models [42, 11, 44]. These static approaches led to rigid coordination [87, 45], limited workflow flexibility [23, 81], and suboptimal agent composition [15, 42, 37, 35]. To address these issues, more adaptive orchestration methods have emerged: network-style organizations dynamically select agents (Dylan [36]) as optimizable graphs enable prompt refinement and better cooperation (GPT-Swarm [87], MacNet [45]); and code-based representations allow modeling of dynamic, task-specific processes. Recent approaches employ code-space search (ADAS [23], AFlow [81]) or train LLMs to generate MAS configurations on demand (MAS-GPT [77]). Beyond LLM-based MAS, classical MARL (Multi Agent Reinforcement Learning) works [26, 65, 34, 63] have long explored coordination and role specialization, offering key inspirations for our RL-driven orchestration in LLM-based multi-agent systems.

## 5  Conclusion

We proposed a novel framework for adaptive multi-agent LLM collaboration inspired by puppet show orchestration, where a centralized, learnable "puppeteer" orchestrator dynamically activated agents within a directed graph-of-thoughts. Unlike previous methods with static or manually designed topologies, our approach incorporated context-sensitive orchestration and reinforcement learning-driven policy adaptation, enabling more principled and efficient collaboration. Experiments on diverse tasks showed that our method achieved superior solution quality and reduced computational cost. Analyses further revealed that the orchestrator promoted compact, cyclic reasoning structures, underpinning the performance improvements. We hope this work can constitute a valuable step toward dynamic and scalable coordination in multi-agent collaboration.

## Acknowledgement

The work was supported by the Tencent Rhino-Bird Focused Research Program.

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

**Appendix**

This appendix complements the main paper by presenting extended evaluation results, detailed implementation configurations of the agents, and a discussion of current limitations.

# A    Supplementary Evaluation

## A.1    Token Cost Analysis

As discussed in Section 3.2, we present here the full set of cost-related results under both the mono-agent and multi-agent settings, covering the Titan and Mimas subspaces. While the main text only reports the results for the multi setting on the Titan subspace, we provide here the remaining three sets of results for completeness: (1) mono on Titan, (2) mono on Mimas, and (3) multi on Mimas. These supplementary results allow for a more comprehensive comparison across different agent configurations and subspace settings. As discussed in Section 3.2, we focus on two key metrics: token consumption and the number of orchestrated agents per reasoning trajectory. While the multi setting has already been analyzed in the main paper, here we focus on the mono setting.

In the Titan subspace, the observed reduction in token consumption across most tasks in Figure 8 can be primarily attributed to shorter reasoning paths. However, for GSM-hard tasks, the number of agents involved remains relatively stable. This suggests that the cost reduction may stem not from fewer agents but from selecting agents that are prompted with fewer tokens, or from utilizing more concise prompts for the same agents. In the Mimas space, token consumption in Figure 8 decreases for certain tasks, while for others, no consistent downward trend is observed. This can be attributed to the fact that, under the mono-agent setting, all agents share the same base model, making it more challenging to identify agents that are significantly more token-efficient.

## A.2    Performance–Cost Trade-off Improvement

To demonstrate the overall improvement in reasoning performance along with the associated reduction in token consumption—both attributed to the effectiveness of our reward design—we track the $\frac{\text{performance}}{\text{cost}}$ score for each sample throughout the entire optimization process. This score is defined as the ratio between task performance (*e.g.,* accuracy or success rate) and token cost, serving as a comprehensive indicator that jointly reflects solution quality and reasoning efficiency.

In the evaluation presented in the main paper, we divide the optimization trajectory into two phases: the initial phase and the evolved phase. This division is introduced solely for evaluation convenience, and does not correspond to any intrinsic change in model behavior. In practice, the online optimization process exhibits a generally monotonic upward trend in $\frac{\text{performance}}{\text{cost}}$, apart from minor fluctuations caused by the stochastic nature of model generation. This consistent improvement further substantiates the effectiveness of our reinforcement learning approach and the benefits of our reward shaping strategy under both mono-agent and multi-agent settings.

Across most tasks, from Figure 9, we observe a clear upward trend in the performance/cost score, indicating that the optimization process leads to both better solutions and more efficient use of tokens. This improvement reflects the ability of the system to identify more suitable agent organizations and reasoning pathways over time. In some cases, the score increases steadily throughout the process, suggesting a smooth convergence toward optimal or near-optimal multi-agent coordination strategies. In other cases, the score initially drops before rising again, which implies that the optimization process explores some less effective multi-agent configurations in early stages but is ultimately able to recover and identify improved structures through continued exploration. This pattern highlights the importance of allowing sufficient exploration in the early optimization phase.

However, for a small subset of tasks, the score does not show a significant upward trend throughout the optimization. We consider two possible reasons for this. First, the number of optimization samples used in our current experiments is limited to 200, which may not be sufficient to fully explore the search space or discover high-quality MAS configurations, especially for tasks with more complex reasoning requirements. Second, the current hyperparameter settings, such as the maximum length of agent sequence allowed, may not be well-suited for tasks involving weaker base models. In such cases, achieving high performance may require larger MAS configurations or higher upper bounds on relevant parameters to allow for more extensive coordination and tool usage.

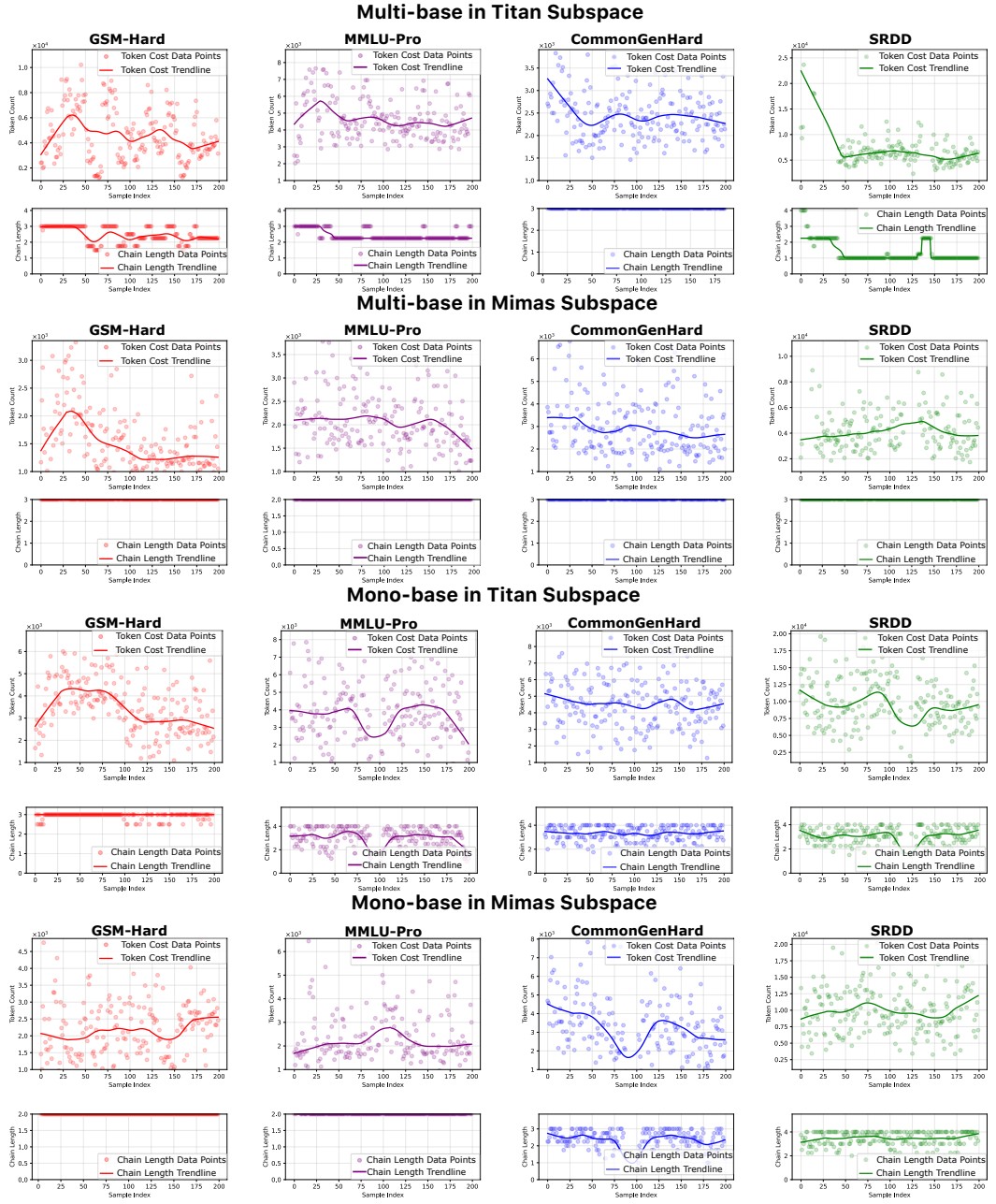

Figure 8: Evolution of token consumption and the number of orchestrated agents per task for all settings. Trends are fitted using LOWESS (Locally Weighted Scatterplot Smoothing).

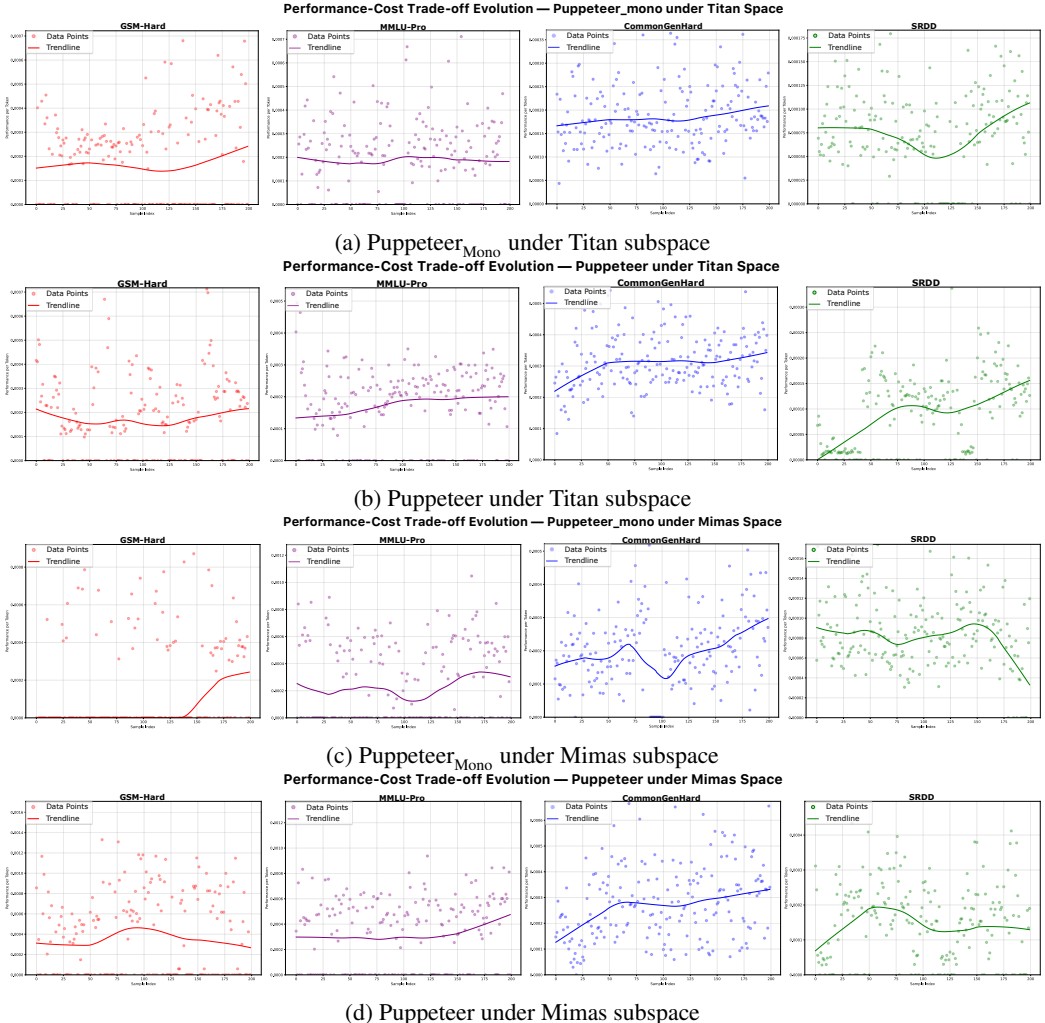

(a) Puppeteer$_{Mono}$ under Titan subspace

(b) Puppeteer under Titan subspace

(c) Puppeteer$_{Mono}$ under Mimas subspace

(d) Puppeteer under Mimas subspace

Figure 9: Performance–Cost curves representing the ratio between task performance and token cost per agent chain across different subspaces and variants of `Puppeteer`. Trend lines are smoothed using LOWESS (Locally Weighted Scatterplot Smoothing).

These observations suggest that while our reward design provides strong guidance for optimizing agent behavior and token efficiency, further gains may be achieved by increasing the optimization budget or dynamically adapting hyperparameters based on task difficulty or model capability.

## A.3 Emergent Patterns in MAS Behavior

To further illustrate the behavioral diversity and coordination capabilities of our multi-agent system (MAS), we present visualizations of several representative MAS trajectories guided by the puppeteer. As shown in the figure, these trajectories reveal a variety of emergent patterns, including both previously validated structures and novel organizational behaviors that arise from agent interactions.

The visualizations in Figure 10 highlight how the policy organizes into distinct formations and adapt agents dynamically in response to high-level guidance. These patterns not only reflect the effectiveness of the puppeteer in orchestrating agent collaboration, but also demonstrate the capacity of the system to generalize and generate new behaviors beyond those explicitly encoded in the training process. Such structural emergence further underscores the interpretability and robustness of our MAS design.

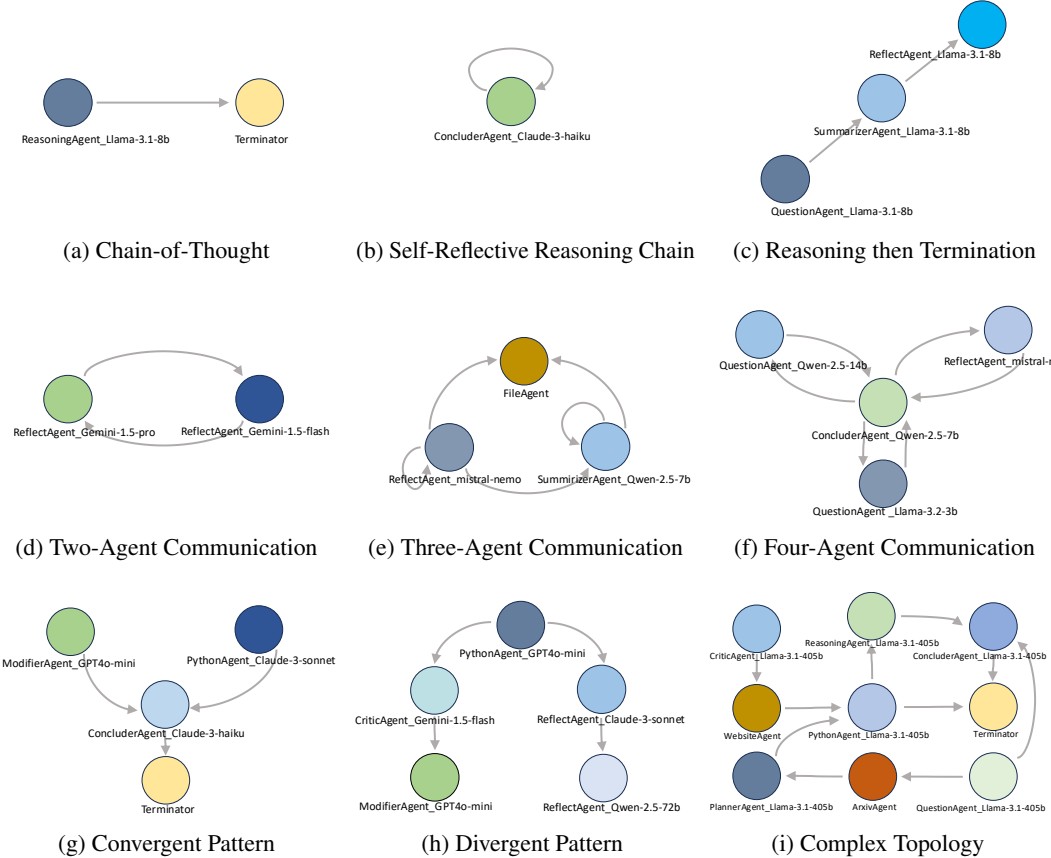

Figure 10: Examples of MAS behavior patterns under the puppeteer's guidance.

## A.4 Generalization to Embodied Environments

While our main experiments focus on tasks such as reasoning, coding, and writing—which do not involve interaction with external environments—we further demonstrate the applicability of our framework to embodied tasks. These tasks require agents to interact with dynamic environments and perform sequences of actions based on real-time feedback, thus posing additional challenges in planning, reasoning, and decision-making.

To showcase this capability, we employ ALFWorld [56] as the testbed. ALFWorld integrates a simulated embodied environment with natural language interfaces, making it a suitable benchmark that bridges textual reasoning with embodied interaction. In this setting, we configure the agent chain with a maximum length of 50, as embodied tasks typically require multiple steps to complete. We also limit the number of exploration trajectories to 1, due to environment constraints: execution cannot be parallelized, and only one admissible action is allowed per step. This experiment is conducted under the mono setting, where all agents share the same model backbone (GPT-4o-mini).

To illustrate how our system operates in embodied settings, we provide a case study visualization using frames saved from the THOR environment (as shown in Figure 11). This example showcases the step-by-step execution of a single task by the agent chain. The successful application of our framework in this setting not only underscores its flexibility across diverse task types, but also demonstrates its extensibility to interactive environments, confirming its potential for integration into broader embodied agent systems that require real-time perception, planning, and execution.

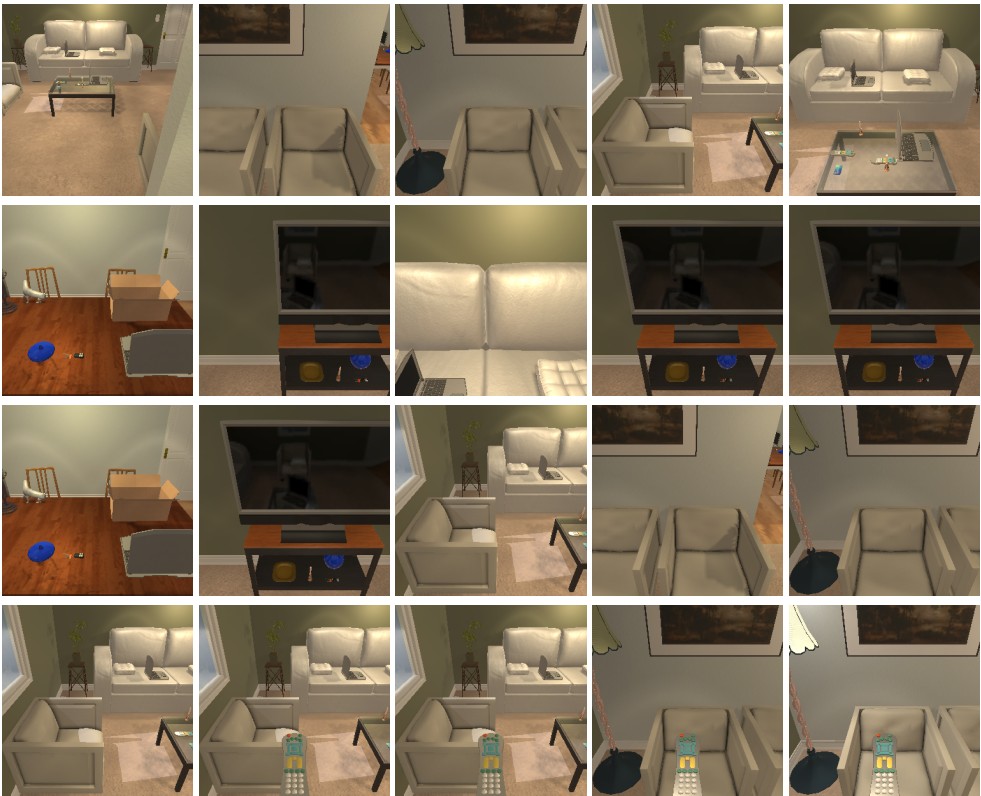

Figure 11: The figures illustrate the implementation process of the task "look for the remote control under the floor lamp."

# B Agent Implementation Details

## B.1 Agent Configuration

We organize all agent actions into two categories: **Tool-use Agent** [47, 85, 51] and **Reasoning Agent** [21, 43] as seen in Table 2. For tool-use agents, their actions involve interacting with external environments, such as querying the web or invoking a code interpreter. In contrast, reasoning patterns reflect the internal cognitive processes of LLMs. Drawing inspiration from prior studies on human reasoning strategies [43, 7, 27, 21], we revisit how humans approach complex reasoning tasks. In practice, individuals adopt diverse strategies: some decompose problems into sub-questions, others solve them directly, and some reframe the problem to emphasize critical constraints. More importantly, humans dynamically adapt their strategies based on the evolving context and problem state. Motivated by these observations, we accordingly design a set of specialized reasoning agents to emulate diverse human strategies. Each agent is equipped with a distinct cognitive role, contributing to a collaborative and adaptive reasoning process.

## B.2 Agent Prompts

In a multi-agent system, to motivate each agent to act in the desired manner, we define its role through a role prompt and combine it with an action prompt to guide the agent in using tools or following specific reasoning patterns. As inllustrated in Fig 12 and Fig 13, role-conditioned prompts specify the agent's identity, domain expertise, and intended responsibilities, guiding the behavior of different agents. And we have designed a set of structured prompts to guide the behavior of our prompt-based agent system, which can be categorized into two major types: tool-augmented action prompts and reasoning-mode prompts. Each prompt specifies a well-defined agent action, accompanied by a template for language model invocation.

Table 2: Agent Tool-Use and Reasoning Patterns

| Category | Name |
|----------|------|
| **Tool-Use** | `read_file`
`search_arxiv`
`search_bing`
`access_website`
`run_python` |
| **Reasoning Patterns** | `reasoning`
`critique`
`reflect`
`question`
`summarize`
`conclude`
`modify`
`planning` |

**Tool-augmented action prompts** in Fig 14 are designed to determine the parameters required for invoking external tools through the agent's environment interface. The goal of these prompts is not to execute the tool directly but to generate structured output that specifies the parameters for tool invocation. For example, such prompts may guide the generation of a search query (*e.g.,* a paper title or keyword) when retrieving academic papers from arXiv `search_arxiv`, or determine the URL to be accessed when invoking a web tool `access_website`. These prompts enable the agent to interface with external systems in a modular and verifiable way by generating the precise input needed to trigger tool execution.

**Reasoning-patterns prompts** in Fig 15 and Fig 16 focus on internal cognitive processes, such as planning, reasoning, critique, reflection, or sub-question generation. These prompts do not rely on external tools but activate structured thinking patterns within the model, enabling complex problem-solving and decision-making through purely generative means.

## C    Computational Resources

| Resource | Specification |
|----------|---------------|
| GPU | NVIDIA A800, 8 GPUs used |
| Peak GPU Memory Usage | 28.8–78.4 GB per GPU |
| Training Time | 2–6 hours |
| Variation Source | Benchmark differences and task complexity |

Table 3: Computational resources used for orchestrator training.

The table (Table 3) details the average computational resources used for orchestrator training. It is important to note that accurately measuring training costs is non-trivial. Our orchestrator is trained online, with parameter updates interleaved with multi-agent inference, making it difficult to isolate GPU hours solely for training. Moreover, several baselines (e.g., AFlow) rely on LLM-based inference-time search to construct workflows, whereas our method uses online gradient-based optimization. Since these involve fundamentally different resource modalities, direct comparisons of computational cost are not appropriate.

## D    Limitations

Our work introduces a centralized mechanism, the Puppeteer, to organize and optimize multi-agent systems (MAS), progressively improving reasoning performance, and efficiency. However, several limitations remain. The Puppeteer's optimization currently depends on coarse-grained rewards based only on final outputs and token usage, lacking informative intermediate feedback. Incorporating fine-

grained supervision, such as step-level correctness, could enhance optimization efficiency. Moreover, the framework assumes a fixed set of agents and tools, limiting adaptability and responsiveness to task variations. Enabling dynamic agent or tool modification during inference would improve flexibility and robustness. Finally, occasional mis-coordination or deceptive agreement among agents suggests the need for more robust interaction protocols and incentive designs. Future efforts may focus on reward shaping and adaptive mechanisms to refine both orchestration and agent-level behaviors, allowing the Puppeteer to make more context-aware and efficient decisions.

Figure 12: Tool-use Agent Role Prompts

---
**FileAgent**

**FileAgent:** You are an expert in file handling. Responsible for reading files and extracting relevant information (`read_file`).

---
**ArxivAgent**

**ArxivAgent:** You are an expert in academic research. Responsible for searching relevant papers on arXiv (`search_arxiv`).

---
**BingAgent**

**BingAgent:** You are an expert in web search. Responsible for retrieving information via Bing (`search_bing`).

---
**WebsiteAgent**

**WebsiteAgent:** You are an expert in accessing and parsing websites. Responsible for extracting data from specific URLs (`access_website`).

---
**PythonAgent**

**PythonAgent:** You are an expert in Python programming. Responsible for executing Python code and returning results (`run_python`).

Figure 13: Reasoning Agent Role Prompts

---

**PlannerAgent**

**PlannerAgent:** You are an expert in task decomposition and planning. Responsible for generating structured plans to solve complex tasks (`planning`).

---

**ReasoningAgent**

**ReasoningAgent:** You are an expert in logical reasoning. Responsible for synthesizing solutions to sub-problems (`reasoning`).

---

**CriticAgent**

**CriticAgent:** You are an expert in critique and verification. Responsible for identifying flaws and providing feedback on prior reasoning (`critique`).

---

**ReflectAgent**

**ReflectAgent:** You are an expert in metacognitive reflection. Responsible for analyzing the overall reasoning trajectory and proposing improvements (`reflect`).

---

**QuestionAgent**

**QuestionAgent:** You are an expert in problem decomposition. Responsible for generating clarifying or follow-up sub-questions (`question`).

---

**SummarizerAgent**

**SummarizerAgent:** You are an expert in summarization. Responsible for generating concise summaries of intermediate results (`summarize`).

---

**ConcluderAgent**

**ConcluderAgent:** You are an expert in synthesis. Responsible for producing the final conclusions based on collective reasoning outcomes (`conclude`).

---

**ModifierAgent**

**ModifierAgent:** You are an expert in error analysis and correction. Responsible for identifying errors and revising prior outputs accordingly (`modify`).

Figure 14: Tool-use Prompts

---

**search_arxiv**

You have chosen to search for academic papers on arXiv. Please provide specific terms related to academic research, such as the title of a paper, keywords, or topics in fields like physics, mathematics, computer science, or machine learning. Return in json format. Example: {"action": "search_arxiv", "parameter": "quantum computing"}

---

**search_wiki**

You have chosen to search for information on Wikipedia. Please provide specific terms like a concept, name, event, or technical term for best results. Return in json format. Example: {"action": "search_wiki", "parameter": "Albert Einstein"}

---

**search_bing**

You have chosen to search for information using Bing. Please provide descriptive phrases or keywords related to your query, including concepts, names, events, or specific questions to get a broad range of results, including news, articles, and websites. Return in json format. Example: {"action": "search_bing", "parameter": "latest advancements in AI"}

---

**access_website**

You have chosen to access a website. Please provide the URL you want to access or the URL most relevant to the current question. Return in json format. Example: {"action": "access_website", "parameter": "https://www.example.com"}

---

**run_python**

You have chosen to write and run Python code. Please write generic Python code in the parameter to solve this type of problems using only standard python libraries. Make sure you use the print function for all output when relevant. Return in json format. Example: {"action": "run_python", "parameter": "print('Hello, World!')"}

---

**read_file**

You have chosen to read a file. Please provide the filename you want to read. Return in json format. Example: {"action": "read_file", "parameter": "data.txt"}

**planning**

Decompose the question and plan the next steps to address the question. You should complete your planning using the following template:
```
REASONING RESULT: [YOUR REASONING RESULT]. *Your previous reasoning was:
{}.*
```
Your planning should include:

**reasoning**

Now, you need to continue the reasoning to get closer to the correct answer. You should finish your reasoning with the following template:
```
REASONING RESULT: [YOUR REASONING RESULT].
```
Finish your answer with:
```
FINAL ANSWER: [YOUR FINAL ANSWER]. *Your previous reasoning was: {}.*
```
You need to follow the direction of the reasoning path and go forward:

**critique**

You need to critique the previous reasoning. Complete your reasoning using:
```
REASONING RESULT: [YOUR REASONING RESULT].
```
Conclude with:
```
FINAL ANSWER: [YOUR FINAL ANSWER]. *Your previous reasoning was: {}.*
```
Consider the following when critiquing: 1. Plausibility:

**reflect**

You will be provided with a previous reasoning attempt where you had access to relevant context and were tasked with answering a question. The attempt was unsuccessful either due to an incorrect answer or a phrasing mismatch with the answer key.
In a few sentences, diagnose the potential cause of failure or discrepancy, and outline a new, concise, high-level plan to prevent the same issue. Use complete sentences.
Reflect on the current state of the task and propose the next steps.
Conclude with:
```
REASONING RESULT: [YOUR REASONING RESULT].
FINAL ANSWER: [YOUR FINAL ANSWER]. *Your previous reasoning was: {}.*
```

**question**

Your task is to propose the next sub-question along with its answer. Ensure it logically follows from the previous reasoning and addresses any gaps.
Provide a well-reasoned answer, supported by evidence or logical arguments.
Conclude with:
```
REASONING RESULT: [YOUR REASONING RESULT].
FINAL ANSWER: [YOUR FINAL ANSWER]. *Your previous reasoning was: {}.*
```

Figure 16: Reasoning-pattern Prompts (Part 2)

**summarize**

You need to summarize previous results and provide some intermediate conclusions.
Finish your reasoning with:
```
REASONING RESULT: [YOUR REASONING RESULT].
```
Then:
```
FINAL ANSWER: [YOUR FINAL ANSWER]. *Your previous reasoning was: {}.*
```
Summarize the reasoning paths and provide a final conclusion.

**conclude**

You need to conclude the task and provide a final answer.
Finish with:
```
REASONING RESULT: [YOUR REASONING RESULT].
```
Then:
```
FINAL ANSWER: [YOUR FINAL ANSWER]. *Your previous reasoning was: {}.*
```

**modify**

You need to identify and correct errors in the previous reasoning.
Use this template:
```
REASONING RESULT: [Clearly state: 1. Which part of the previous reasoning
was incorrect 2. Why it was incorrect 3. What is the correct understanding].
```
Then:
```
FINAL ANSWER: [Provide the complete corrected answer].  *Your previous
reasoning was: {}.*
```
Please explicitly point out and correct any errors, misconceptions, or inaccuracies.

