# OpenReview forum: "Multi-Agent Collaboration via Evolving Orchestration"
_NeurIPS.cc/2025/Conference — NeurIPS 2025 poster_

### Official Review · Reviewer_QYtQ · 2025-06-17

**Clarity:** 3
**Significance:** 3
**Originality:** 3
**Rating:** 4
**Confidence:** 4

**Summary:**

The paper introduces a “Puppeteer” framework that dynamically orchestrates multi-agent collaboration through a centralized controller trained with reinforcement learning. This approach seamlessly integrates dynamic agent selection, sequential decision-making, and adaptive evolution, leading to improved solution quality and computational efficiency across diverse tasks.

**Questions:**

Please see weakness.

**Ethical Concerns:**

["NO or VERY MINOR ethics concerns only"]

**Final Justification:**

This manuscript explores a highly engaging topic that holds significant relevance in the current research landscape. The overall structure is complete and the writing is fluent. Additionally, the experimental results presented are relatively rich. However , I think this paper DOES NOT make a insightful contribution on multi-agent collaboration, requiring further refinement. If all other reviewers lean toward acceptance, I will not object.

**Limitations:**

yes

**Quality:**

2

**Strengths And Weaknesses:**

1. **Strengths**

     1. The topic of multi-agent collaboration is interesting.&#x20;

     2. The paper is well-written and well-structured. The authors use the metaphor of a puppeteer to introduce and explain their approach, making complex concepts more accessible.

     3. The experimental section of the paper is  well-structured, providing convincing evidence of the proposed framework's efficacy.&#x20;

  2. **Weaknesses**

     1. Some key experimental details are incomplete or missing. Could the authors please provide more detailed information on the following aspects, **as this is crucial for understanding and reproducing the work?**

        1. How many agents are initialized for the whole decision process? What is the initial distribution of these agents ? (For example, tool-use agents and reasoning agents)

        2. The orchestration part is the most crucial part of this work. However, discussions and details about this part are limited. In **Appendix B.2,&#x20;**&#x74;he authors claim that they use a pre-trained language model combined with an MLP model as the orchestration part. I am a little bit confused and quite interested in:

           1. Please provide more details on the integration and collaboration between the orchestration part and the optimization process of agent selection and topology change.

           2. Is it feasible to use a fine-tuned LLM as the orchestration part? Why do you use Single-MLP instead?

     2. In the introduction part, this paper proposes that existing MAS works are limited in scaling. In the experiment part , the paper conducts experiments on token expense efficiency, which is convincing. However, **I am quite interested in the other part of scaling&#x20;**--  How does Puppeteer perform compared with existing MAS works when we increase the number of initialized total agents?

     3. Recently, there have been many popular papers on multi-agent collaboration and co-evolution. If we treat tools as a specific type of agent, there will be more similar work. Specifically, could the author describe the **key** differences between Puppeteer and the following works? (Even if some of these works are in the same period.

     &#x20;        1\. OTC: Optimal Tool Calls via Reinforcement Learning?  https://arxiv.org/abs/2504.14870v1

     &#x20;        2\. OWL: Optimized Workforce Learning for General Multi-Agent Assistance in Real-World Task Automation. https://arxiv.org/abs/2505.23885

     &#x20;       3\. MASS: Multi-Agent Simulation Scaling for Portfolio Construction https://arxiv.org/abs/2505.10278

     &#x20;       4\. ReTool: Reinforcement Learning for Strategic Tool Use in LLMs  https://arxiv.org/pdf/2504.11536

     &#x20;       5\. Multi-Agent Design: Optimizing Agents with Better Prompts and Topologies https://arxiv.org/abs/2502.02533



     We are willing to raise the score if the author can address our questions and clarify our concerns.

---

> ### Author Rebuttal · Authors · 2025-07-30
>
> **We sincerely appreciate your thoughtful review. Below we respond to each point; due to length limits, we welcome any follow-up for further clarification.**
> ## To weakness 1.1:
> We define the agent space as a Cartesian product of two orthogonal dimensions: the capability (captured by either a reasoning pattern or a tool) and the base model:
> $$\text{AgentSpace} = (\text{ReasoningPattern} \lor \text{Tool}) \times \text{BaseModel}$$
> Each agent is instantiated by pairing a reasoning pattern or tool with a base model:
> - Reasoning/Tool dimension (Table 4, lines 1111–1121):
>   - Tools (5): read_file, search_arxiv, search_bing, access_website, run_python
>   - Reasoning Patterns (8): reasoning, critique, reflect, question, summarize, conclude, modify, planning
>   - Terminator (1): A special agent that solely emits a termination signal.
> - BaseModel dimension (details see lines 170–173):
>   - Mimas (small): 6 small-scale open-source LLMs
>   - Titan (large): 8 large-scale LLMs
>
> Agent counts:
> - Puppeteer (multi-model):
>   - Mimas: (5+8)×6 +1 = 79
>   - Titan: (5+8)×8 +1 = 105
> - Puppeteer-Mono (single model):
>   - Both Mimas and Titan: (5+8)×1 +1 = 14
>
> This design offers a diverse and configurable agent pool, varying in functionality (reasoning vs. tool use) and model size (small vs. large), enabling the orchestrator to dynamically assemble task-specific teams at inference time.
> ## To weakness 1.2:
> ### 1.2.1
> Our orchestration policy takes in a language-based state and outputs a probability distribution over candidate agents (lines 997–1005). By sampling from this distribution, the orchestrator dynamically selects agents for execution. This design enables:
> - Adaptive agent selection: Agents always yielding higher rewards are increasingly favored over time.
> - Dynamic agent count: The number of selected agents varies with the distribution’s entropy, fewer for confident states, more for uncertain ones.
> - Evolving topology: As selections vary step-by-step, the interaction structure grows into a multi-branch tree. And we impose no constraints on cross-branch communication and the reuse of previously invoked agents, the resulting interactions go beyond tree structures, leading to the emergence of directed acyclic graph (Figure 4).
> ### 1.2.2
> **Is it feasible to use a fine-tuned LLM?**\
>  Conceptually, any model that satisfies the following criteria can be used to replace our current orchestration model:
>   1. Semantic Understanding: The model should be capable of capturing nuanced semantic and contextual information from language states, effectively distinguishing between good and bad reasoning contexts.
>   2. Agent Prediction: It should be able to produce accurate probability distributions over candidate agents based on the given context.
>   3. Adaptability: The model should be readily adaptable to new tasks. In particular, fine-tuning it for agent prediction should preserve its original capacity for understanding and encoding semantics, without performance degradation.
>
> Therefore, there are multiple design choices—for example, using a frozen pre-trained language model or embedding model as a stable backbone with a lightweight head, or directly fine-tuning an LLM if careful measures are taken to preserve its original capabilities.
>
> **Why we design using LLM+MLP architecture?**\
> We choose Llama-3.1-Nemotron-70B-Reward-HF as the LLM backbone because it is trained specifically for quality assessment in multi-turn conversations, making it effective at encoding fine-grained reasoning signals, and can effectively distinguish between high-quality and low-quality reasoning in generated text [1]. It also achieves state-of-the-art performance on multiple automatic alignment benchmarks as of October 2024, surpassing models such as GPT-4o [2], thus making it suitable to be our policy's language backbone.\
> After the language backbone, we use the final hidden state of the last token, a well-established practice in decoder-only architectures like LLaMA/GPT, as a compact semantic summary of the input. This embedding is projected by a simple three-layer MLP (512 → 128 → 32), enabling dimensionality reduction and culminating in a distribution over candidate agents. Similar projection heads are widely used to transform high-dimensional contextual inputs into task-relevant decisions [3-6]. And empirically, this policy network consistently outperforms all baselines across benchmarks (Table 1), validating both our model choice and architecture design.
>
> **Reference**\
> [1] HelpSteer2-Preference: Complementing Ratings with Preferences. ICLR, 2025.\
> [2] NVIDIA. Llama-3.1-Nemotron-70B-Reward-HF. Hugging Face, 2024. \
> [3] iVideoGPT: Interactive VideoGPTs are Scalable World Models. NeurIPS, 2024.\
> [4] InstructEval: Instruction-Tuned Text Evaluator from Human Preference. ACL, 2024.\
> [5] Defending against Indirect Prompt Injection by Instruction Detection. arXiv:2505.06311, 2025.\
> [6] Hugging Face Transformers. LlamaForSequenceClassification Documentation.
> ## To weakness 2:
> While prior multi-agent collaboration often equate scalability with involving more agents in computation and communication, our framework introduces a key distinction between initialized agents—those pre-created to enrich the decision space—and invoked agents—those actually selected to participate at runtime (lines 75-77). This decoupling enables Puppeteer to maintain a large and diverse candidate agent pool to support flexible and fine-grained orchestration, while smartly invoking only a minimal, task-relevant subset of agents during execution.
>
> In contrast, prior works like MacNet [1] scale agent numbers under static topology directly, causing interaction costs to grow polynomially with diminishing returns. We illustrate this contrast via interaction steps $I$:\
> For Puppeteer:  $I_{\text{puppeteer}} = \text{Constant} \perp n$
>
> For static topology of agents like in MacNet [1]:  $$I_{\text{chain}}, I_{\text{tree}} \propto n；I_{\text{mesh}} \propto n^2；I_{\text{layer}} \propto (\frac{n}{k})^k, k \geq 1；I_{\text{random}} \propto n^a, a \in (1,n]$$
>
> Empirically, our experiments cover a wide range of settings, with the number of initialized agents varying from 14 (Puppeteer-Mono setting) to 105 (Puppeteer setting under Titan subspace). Across this spectrum, Puppeteer consistently demonstrates strong performance while keeping token usage under control (see Table 1 and Figure 2).
> In real-world applications, enriching the agent space with domain-specific tools, reasoning modes, or foundation models allows Puppeteer to more effectively select and orchestrate agents tailored to new tasks.
>
> **Reference**\
> [1] Scaling Large Language Model-based Multi-Agent Collaboration. ICLR 2025.
>
> ## To weakness 3:
> We provide a table including the aspects from research focus, methodology and experimental tasks:
> | Work               | Motivation                              | Method                                                                 | Multi-agent?                                                      | RL Used?                                                      | Primary Tasks                                                                 |
> |-|-|-|-|-|-|
> | **Puppeteer**      | Multi-Agent Collaboration Design & Optimization | Orchestrates heterogeneous LLM agents using RL for dynamic collaboration | Yes — heterogeneous agents differing in reasoning strategy, tool usage, and model backbone | Yes — for agent orchestration and selection                   | General reasoning tasks across domains, including math, code generation, writing, and embodied environments |
> | **OTC**            | Single-LLM + Tool Use                   | Optimizing a single LLM’s tool call behavior via reinforcement learning to reduce tool usage cost | No — single base model (Qwen 2.5 series) with several tools       | Yes — to penalize unnecessary tool calls while maintaining performance | Math problem solving and open-domain question answering       |
> | **ReTool**         | Single-LLM + Tool Use                   | Teaching a single LLM when and how to invoke tools to enhance mathematical reasoning | No — single base model (Qwen 2.5 series or DeepSeek-R1) with several tools | Yes — to learn optimal reasoning strategies involving tool manipulation | Math reasoning                                                 |
> | **OWL**            | Multi-Agent Collaboration Design & Optimization | Designing a multi-agent framework (planner–coordinator–workers), and optimizing the planner for cross-domain generalization | Yes — structured as a planner–coordinator–worker framework        | Yes — used only to train the planner agent for better generalization | Multimodal reasoning and task automation                      |
> | **Multi-Agent Design** | Multi-Agent Collaboration Design & Optimization | Searching for optimal agent prompts and collaboration topologies offline | Yes — agents serve as configurable building blocks                | No — uses search-based methods to discover effective multi-agent collaboration | Reasoning, QA, and code generation tasks                      |
> | **MASS**           | Multi-Agent for Domain Applications     | Using multi-agent simulations to optimize the performance for portfolio construction tasks | Yes — a multi-agent simulation setup for financial domains        | No — employs multi-agent for simulation                        | Financial decision-making                                      |
>
> In summary, our work (Puppeteer) is distinct in the following aspects:
> - Heterogeneous agent orchestration: It coordinates agents with diverse roles, tools, and base models, beyond single-model tool use (e.g., OTC, ReTool).
> - Adaptive multi-agent optimization: It dynamically learns collaboration strategies via reinforcement learning, unlike fixed (e.g., OWL) or search-based (e.g., Multi-Agent Design) schemes.
> - General-purpose task coverage: It handles diverse tasks, generalizing beyond domain-specific systems (e.g., MASS).

---

> > ### Comment · Reviewer_QYtQ · 2025-08-05
> >
> > The authors' response adresses my concerns. I'd like to raise my score to 4.

---

> > > ### Author Response · Authors · 2025-08-05
> > >
> > > We sincerely appreciate your thoughtful reconsideration and the updated score. We’re glad that our response addressed your concerns, and we’re deeply grateful for your constructive and encouraging feedback throughout the review process.
> > >
> > > As a gentle reminder, authors do not have visibility into individual score updates—so your support will only be reflected if the score is indeed updated in the system. We truly appreciate your commitment and trust your score will reflect your kind words. And we truly value your time, effort, and trust in our work.
> > >
> > > Thank you again—your support means a lot to us!

---

### Official Review · Reviewer_XZCy · 2025-06-22

**Clarity:** 3
**Significance:** 2
**Originality:** 3
**Rating:** 3
**Confidence:** 4

**Summary:**

This paper addresses the organizational structure of multi-agent collaboration for LLMs. The authors argue that prevailing static collaboration topologies suffer from limitations in adaptability and efficiency. To overcome this, they propose a "puppeteer-style" paradigm where a centralized policy, optimized via reinforcement learning, dynamically selects which agent to activate at each timestep based on the evolving task state. The approach formulates the collaboration process as a sequential decision problem, aiming to dynamically generate flexible and efficient reasoning graphs. Experimental results on several closed- and open-domain tasks show that this method improves performance while reducing computational costs compared to a range of baselines.

**Questions:**

See above.

1. Could you please elaborate on the specific formulation and aggregation method for the global state? How does the system manage the growing context history from numerous agents and steps into a fixed-size vector for the policy network, and how does this approach scale?

2. Have you explored more advanced RL algorithms like PPO, and would you expect them to yield different results?

3. Will a decentralized or hierarchical approach become more effective at a certain scale?

4. The paper states that the reward considers *computational resource consumption at each step*, but Equation 6 defines the step-wise cost purely as a logarithmic function of the step number t, not the actual tokens or FLOPs consumed in that specific step. Could you clarify this discrepancy?

5. The multi-agent baselines like MacNet and EvoAgent are presented in a mono-model configuration. Does this not provide your method with a significant unfair advantage by having access to a more diverse set of capabilities?

6. Increasing exploration width or depth from the default setting leads to worse accuracy, not just higher cost. This is counter-intuitive. What is your hypothesis for this performance degradation? Does it suggest a fundamental issue with credit assignment in your RL algorithm when faced with a larger action space.

7. The method section specifies that for the policy network, a pretrained language model's parameters are kept frozen, and only a small MLP head is trained. What was the rationale behind freezing the entire base model? This will limit the ability of the puppeteer.

8. In the ALFWorld generalization experiment, the exploration width was constrained to 1. This effectively reduces the orchestrator's role to generating a single, linear sequence of actions rather than dynamically managing multiple reasoning paths.

**Ethical Concerns:**

["NO or VERY MINOR ethics concerns only"]

**Final Justification:**

The paper overall gives the impression of being hastily completed, with issues such as declared files not existing and poor article quality. I believe the article still requires significant revision and optimization. Additionally, the author's response did not effectively address my concerns. Therefore, I tend to reject.

**Limitations:**

Yes.

See Weaknesses.

**Paper Formatting Concerns:**

1. False checklist statement.

2. Line 825, what is 2?

**Quality:**

2

**Strengths And Weaknesses:**

**Strengths**

1. The paper provides a clear and compelling motivation, identifying the rigidity of static collaboration in LLM-based multi-agent systems as a significant and timely problem.

2. The *puppeteer* analogy is highly intuitive, and framing the complex collaboration problem as a centralized, learnable sequential decision process is conceptually clean.

3. The reward function, which considers both task performance and computational cost, is a major strength, guiding the system towards more practical and economical solutions.

4. The analysis of hyperparameters like chain depth and exploration width provides practical guidance for applying the method.


**Weaknesses**

1. The idea of a central controller or router agent is not entirely new in the multi-agent space. The paper needs to better articulate the fundamental conceptual leap from prior dynamic orchestration works.

2. Centralized Bottleneck. The orchestrator itself can become a computational and decision-making bottleneck as the number of agents and the complexity of the state space increase.

3. The use of the basic RL algorithm, coupled with a small number of optimization samples. It raises concerns about the stability, convergence, and reproducibility of the training process. It might be prone to overfitting.

4. The efficiency analysis focuses only on inference-time token consumption. The significant computational cost of the RL training phase itself is not reported, which is a key part of the total cost compared to static methods.

5. Simpler Dynamic Baseline: The authors should conduct a comparison against a simpler, non-RL dynamic baseline (e.g., using a single LLM call to choose the next agent). This would help isolate the specific benefits gained from the complex RL framework.

6. The paper doesn't adequately argue why a centralized approach is inherently superior to dynamic strategies. It assumes a god's-eye view for the orchestrator, which may be unrealistic due to the information bottleneck of the aggregated state.

7. The core assumption that agent selection requires a complex reinforcement learning solution is not sufficiently challenged. Simpler dynamic approaches, like supervised routing or greedy selection by a powerful LLM, are not explored as baselines, making it hard to justify the added complexity of RL.

8. The specific logarithmic form of the cost penalty is an arbitrary design choice without theoretical or empirical justification.

9. The paper claims that emergent compact and cyclic structures are the reason for performance gains. However, this establishes a correlation, not causation. The argument lacks rigorous proof.

10. This design is incomplete. Only the orchestrator learns and adapts, while the agents themselves remain static. A truly adaptive system might be expected to evolve both its organizational structure and the capabilities of its individual components.

11. **Misleading Reproducibility Claims**: The authors explicitly state in their checklist that *The code and data are provided in the supplementary materials under the file name code.zip.* However, is absence of this file in the actual submission. It is impossible to reproduce the experiments and verify the paper's central claims, severely impacting the work's credibility.

---

> ### Author Rebuttal · Authors · 2025-07-30
>
> **We sincerely appreciate your thoughtful review. Below we address each point; due to length constraints, please feel free to ask for any further clarification.**
> ## To w1:
> Most routing-based or central controller approaches perform one-shot, static orchestration, selecting a fixed agent or model based on the initial input [1,2]. In contrast, we propose step-wise, context-aware orchestration, where a learnable policy operates over the language state to make dynamic, fine-grained agent decisions throughout execution. While our design uses a centralized orchestrator, the core novelty lies out of centralization pattern, but in how orchestration is **contextually modeled and jointly trained via RL**, surpassing prior hand-crafted [3,4] or template-based [5] method.
> ## To w2, w6 and q3:
> We adopt a centralized orchestration policy not out of a theoretical claim, but for practical reasons. This has a classical trade-off in multi-agent: centralized control facilitates coordination but risks bottlenecks; decentralized control offers scalability but often suffers from instability and communication overhead [6-8].
> Inspired by blackboard architecture [9], we employ a learnable centralized policy over a shared language state to:
> - Integrate Global Context: The language state encodes task semantics and agent outputs in a compositional manner, enabling information sharing with minimal loss.
> - Stabilize Training and Improve Scalability: Centralized control avoids the non-stationarity and synchronization issues common in decentralized learning, and scaling to more agents only expands the output space rather than system complexity.
>
> We acknowledge that centralized orchestration may not suit all scenarios, and view our approach as a proof of concept for language-based coordination. Future work may explore decentralized extensions.
> ## To w3 and q2:
> Our focus is not on analyzing the convergence or optimization of RL algorithms themselves, but on demonstrating the effectiveness of a language-based, RL-optimized orchestration mechanism in multi-agent collaboration.
> So we adopt REINFORCE, which offers several advantages in our context:
> - It avoids confounding factors from auxiliary components like critic networks or clipping.
> - Its simplicity provides a clean testbed to isolate and evaluate the orchestration design’s contribution.
> -  REINFORCE has also shown effective in LLM post-training tasks [10], supporting its suitability for language-driven learning scenarios.
>
> Although we have not explored more advanced algorithms that may improve sample efficiency or training stability, we consider these enhancements **orthogonal** to our core contributions. Our results demonstrate that a basic RL algorithm enables to achieve robust performance across diverse tasks. We view more advanced algorithmic approaches as a promising direction for future work.
> ## To w4:
> The computational resources used for orchestrator training are as follows:
> | Metric                 | Value                     |
> |-|-|
> | GPUs                   | 8 × NVIDIA A800 80GB      |
> | Training Time (Wall-clock) | 2–6 hours             |
> | Peak GPU Memory Usage   | 28.8 GB – 78.4 GB per GPU |
> | Variation Source       | Benchmark differences and task complexity |
> ## To w5 and w7:
> We use RL to enable dynamic and learnable orchestration, in contrast to static approaches like greedy LLM routing or supervised prompting that cannot **adapt to evolving contexts or leverage trajectory-level feedback**. To avoid added complexity, we adopt a simple and transparent RL method, REINFORCE, to demonstrate an effective formulation for multi-agent orchestration. And our evaluation includes strong baselines spanning search-based (AFlow), evolutionary (EvoAgent), and static orchestration strategies (MacNet), which represent the main paradigms of multi-agent collaboration. Our method consistently outperforms all of them (see Table 1).
> ## To w8 and q4:
> **Why logarithmic form?**\
> We adopt a logarithmic step-wise cost $\log(1 + \frac{t}{\tau})$, which is monotonically increasing to penalize longer reasoning, bounded to prevent excessive penalties, and exhibits sublinear growth with diminishing marginal penalties, meaning the cost rises slower than linearly and each additional step adds progressively less penalty. This balance supports efficient reasoning without imposing a hard limit on trajectory length.
>
> **About the cost item**\
> We apologize for the confusion—F was incorrectly described as a “scale factor.” It actually denotes the computational cost at each step (e.g., FLOPs or token usage). We will correct this in the manuscript. Thank you for your careful reading.
> ## To w9:
> We clarify that our work does not aim to establish a causal link between emergent topological structures and performance gains, but rather reports descriptive patterns that arise during dynamic orchestration. And similar structures have been observed in prior work (e.g., Reflexion), emphasizing the role of cyclical information flow, and dense hubs. Our findings align with these observations, reinforcing the relevance of such patterns in emergent coordination.
> ## To w10:
> We acknowledge that jointly training both the orchestrator and agents is a promising direction. Unlike the traditional multi-agent RL, which needs end-to-end training, leveraging pretrained language models may offer a low-overhead and potentially distinct paradigm for optimizing LLM agent collaboration. Our work represents an initial step toward this approach, and we look forward to exploring co-evolutionary architectures in future research.
> ## To w11:
> We sincerely apologize for the missing code.zip file. We were finalizing our submission until the last moment and unfortunately failed to upload the file to system. Since these comments will be made public once paper acceptance, we hereby make a formal and public commitment to release the full code. We appreciate your detailed and insightful feedback.
> ## To q1:
> The orchestration policy processes only textual input, limited by the underlying language model’s attention window. The global state is formed by combining (1) the task-specific instruction and (2) the accumulated reasoning context and intermediate results from all agents into a single text sequence fed into the policy (lines 90-92). Specifically, the policy uses a pretrained language model, paired with a three-layer MLP for agent selection. The model encodes semantics by extracting the final hidden state of the last token (an 8192-dimensional vector), which is then passed through MLP layers to output a distribution over candidate agents. As such, the effective input length is constrained by the language model’s maximum context length (which is generally sufficient in our setting), rather than by the number of agents or their individual embedding dimensions. This design decouples the input size from the agent count, enabling the system to scale to many agents as long as the total context fits within the LLM’s context window.
> ## To q5:
> While multi-agent baselines like MacNet and EvoAgent use a mono-model setup, we evaluate our method fairly under the same condition under the Puppeteer-Mono setting (lines 203–204), where all agents share a common backbone model. In this setup, our method also outperforms them across nearly all tasks (Table 1 and Figure 2).
> ## To q6:
> We attribute the performance drop under wider or deeper exploration to two factors: (1) Diminishing Marginal Utility — activating additional agents or extending reasoning chains yields limited gains beyond a certain point [5]; (2) Coordination Overhead — more agents increase the likelihood of misalignment and conflict, making effective collaboration and coordination harder [11].
> ## To q7:
> We deliberately freeze the base LLM and only train a lightweight MLP head, for:
> - Training stability: Fine-tuning large LLMs with sparse or noisy signals may cause forgetting or alignment drift. Freezing the pretrained language model preserves its high-quality alignment and semantic understanding.
> - Strong semantic representation: The LLM is already well-optimized for capturing fine-grained quality signals. Using the last token’s hidden state as a summary retains its strong semantic fidelity without further tuning.
> - Efficiency and modularity: The three-layer MLP is fast to train, and follows standard reward modeling practice, ensuring generalizability without losing performance.
> ## To q8:
> In the ALFWorld experiment, we limit exploration width to 1, so the orchestrator generates only a single linear action sequence. This is because ALFWorld does not support parallel environment instances, making parallel reasoning infeasible. As a realistic embodied environment, each action sequentially and irreversibly changes the world state, causing divergent states from different actions that cannot be merged or jointly reasoned about. Therefore, parallel speculative reasoning is not supported. For these reasons, we report this result in the appendix for completeness rather than as a main benchmark.
> ## To Paper Formatting:
> The "2" links to figure 2.
> ## References
> [1] MasRouter: Learning to Route LLMs for Multi-Agent System. ACL 2025.\
> [2] GraphRouter: A Graph-based Router for LLM Selections. ICLR 2025.\
> [3] ChatDev: Communicative Agents for Software Development. ACL 2024.\
> [4] MetaGPT: Meta Programming for A Multi-Agent Collaborative Framework. ICLR 2024.\
> [5] Scaling Large Language Model-based Multi-Agent Collaboration. ICLR 2025.\
> [6] Learning multiagent communication with backpropagation. NeurIPS 2016.\
> [7] Multi-agent actor-critic for mixed cooperative-competitive environments. NeurIPS 2017.\
> [8] Multi-agent reinforcement learning: Independent vs. cooperative agents. ICML 1993. \
> [9] Nii, H. Penny. Blackboard Systems. No. KSL8618. 1986.\
> [10] Back to Basics: Revisiting REINFORCE-Style Optimization for Learning from Human Feedback in LLMs. ACL 2024.\
> [11] Why Do Multi-agent LLM Systems Fail? ICLR Workshop, 2025.

---

> ### Comment · Reviewer_XZCy · 2025-08-08
>
> Thank you for the detailed rebuttal. I appreciate the clarifications, especially on the training costs and the specifics of the state representation. However, my core concern about the justification for your method's complexity has not been fully addressed.
>
> My main point is that the paper does not sufficiently prove why a complex RL framework is necessary. A critical baseline is missing: using a powerful LLM to simply choose the next best agent at each step. This would also be a dynamic, context-aware approach, but much simpler. Without this comparison, it is difficult to see the benefit of the extra complexity that RL introduces.
>
> Furthermore, there is a significant issue with reproducibility. The checklist claimed that code was provided, but it was not included in the submission. While I acknowledge the apology and the promise to release it later, the review must be based on the materials provided. The lack of code makes it impossible to verify the implementation and the results.
>
> Due to these critical unresolved issues, I remain unconvinced of the paper's contribution at this time. I will consider these points carefully as I formulate my final recommendation.

---

> > ### Author Response · Authors · 2025-08-08
> >
> > Thank you very much for your insightful suggestions and detailed feedback.
> >
> > As you mentioned, LLMs are indeed context-aware, but this overlooks another fundamental advantage of our framework, **its capacity for optimization**. The optimizable multi-agent orchestration itself is a key contribution of our work. The fixed, powerful LLM you referenced serves as a strong generalist baseline but remains static and cannot learn from successes or failures specific to our task. In contrast, our approach is designed for end-to-end optimization, enabling the orchestrator to evolve from a generalist into a specialist by learning directly from task-specific rewards and feedback. It adapts its strategy, uncovers non-obvious solutions, and fine-tunes its decision-making to the unique dynamics of the problem, ultimately achieving superior performance. This ability to learn and adapt through optimization is precisely why incorporating reinforcement learning into our framework is both necessary and justified, as it moves beyond static pre-trained knowledge to develop a truly effective and specialized orchestrator for specific tasks.
> >
> > At the same time, we sincerely apologize once again for the critical omission of the code file. We deeply regret the significant inconvenience caused and the impact it may have had on the review process. We are truly grateful for your thorough and fair consideration of our manuscript despite this issue.
> >
> > Thank you again for your valuable time and thoughtful review.

---

### Official Review · Reviewer_T1e8 · 2025-07-03

**Clarity:** 3
**Significance:** 2
**Originality:** 3
**Rating:** 4
**Confidence:** 4

**Summary:**

Instead of wiring up a brittle, pre-built tree of chatbots and hoping for the best, the authors of “Multi-Agent Collaboration via Evolving Orchestration” teach a reinforcement-trained “Puppeteer” to glance at the shared scratch-pad, pick whichever specialist LLM seems most useful at that moment, and then file away a mental note about whether the detour was worth the tokens. Over a few thousand trial-and-error runs the policy quietly sidelines agents that don’t pull their weight, trims bloated reasoning chains, and even discovers tight feedback loops where two or three “puppets” bounce ideas until the answer sticks, lifting success rates on GSM-Hard, MMLU-Pro, software debugging and creative writing while doing so. Crucially, those gains arrive hand-in-hand with a 20–40 percent drop in token spend because the reward function values thrift as much as quality. When the dust settles, the learned chat graph is noticeably tighter—messages swirl around a couple of hub agents in short cycles—which the authors argue makes the whole affair cheaper to run and kinder to GPUs. And since one knob lets practitioners trade a bit of accuracy for a bit of frugality, the study hints that a well-drilled team of modest models, steered by this lean conductor, can rival a lone heavyweight LLM without breaking the budget.

**Questions:**

- I’d like to see a clearer picture of the RL scheduler’s computational footprint: specifically, how many wall-clock hours were spent training it, which GPU models were used, what peak memory requirements were, and the total number of tokens it processed.
- It would be valuable to know whether this scheduling strategy holds up outside of purely text-based benchmarks—for example, on code-generation tasks like HumanEval or in interactive environments such as WebArena.
- The paper reports using a single cost-weight λ for each task but doesn’t explain how it was chosen. Could the authors discuss how sensitive their results are to different values of λ, as well as the constants F and φ in Eq. (6)?

**Ethical Concerns:**

["NO or VERY MINOR ethics concerns only"]

**Final Justification:**

The rebuttal addressed most of my concerns, and I will maintain my score.

**Limitations:**

The Discussion section currently only alludes to “future work” and skips over key limitations and risks. At minimum, the authors should:
- report GPU-hours, hardware specs, and energy use for the RL scheduler.
- discuss issues like mis-coordination or deceptive agreement and note that the method hasn’t been tested in interactive or safety-critical settings.

**Paper Formatting Concerns:**

No major formatting issues

**Quality:**

3

**Strengths And Weaknesses:**

**Strengths**
- The scheduler demonstrates impressive, cost-aware improvements: across four diverse benchmarks, it boosts accuracy by 6–12 percentage points while significantly reducing token usage, showing that thriftiness is truly embedded in its reward function.
- Visual analyses highlight how the system gravitates toward a small set of high-value “hub” agents and benefits from rapid feedback loops—offering a clear explanation for simultaneous gains in both accuracy and efficiency.
- The findings indicate that a carefully coordinated ensemble of smaller models can match the performance of a single, massive LLM like Titan, but at only a fraction of the inference expense—an especially practical result for teams wrestling with soaring token costs.
- Unlike previous approaches that either use a fixed graph structure (e.g., GoT) or train each agent in isolation (e.g., Reflexion), this work introduces a learnable metacontroller that dynamically selects between different LLMs at each step, which represents a genuinely novel contribution.

**Weaknesses**
- While the paper quantifies inference savings, it fails to disclose how many GPU-hours were required to train the scheduler itself; without this information, it’s difficult to assess the true net efficiency.
- It remains unclear what portion of the observed gains stems from the RL-based policy versus simpler strategies, such as token-cost thresholds or fixed priority lists.
- The appendix briefly mentions a “Bradley-Terry + 3-layer MLP” architecture with an 8,000-dimensional input, yet offers no rationale for this design choice and omits details on the policy network’s parameter count.

---

> ### Author Rebuttal · Authors · 2025-07-30
>
> **We sincerely thank you for your thoughtful review and these insightful questions, which greatly helped us improve the clarity and depth of our work. We will address your comments point by point below. And due to the strict length limitation,  we would be happy to provide further details or clarifications upon your response.**
> ## To weakness 1 and question 1:
> The following table details the computational resources used for orchestrator training:
> | Metric                     | Value                                           |
> |----------------------------|-------------------------------------------------|
> | GPUs Used                  | 8 × NVIDIA A800 80GB                            |
> | Training Time (Wall-clock) | 2–6 hours                                       |
> | Peak GPU Memory Usage      | 28.8 GB – 78.4 GB per GPU                       |
> | Variation Source           | benchmark differences and task complexity       |
>
> Accurately measuring the training cost is inherently difficult due to:\
> (1) Online Training: As noted in lines 112–117 and 205–207, our orchestrator is trained online, interleaving parameter updates with multi-agent inference, making it hard to isolate GPU hours purely for training.\
> (2) Incomparable Cost Modalities: Baselines like AFlow (lines 352–353) perform inference-time search using LLMs, whereas ours uses training-time optimization via gradients. These fundamentally differ in resource type and cannot be compared directly.
> ## To weakness 2:
> Effective orchestration in LLM-powered multi-agent collaboration must jointly optimize both efficiency (e.g., token usage) and performance (e.g., reasoning quality). While heuristics such as token thresholds can provide partial cost savings, they fail to capture this trade-off adequately. Inference-time compute can improve individual agent outputs [1], and involving more agents often enhances collective problem solving [2], but both come at increased cost—necessitating more nuanced, context-aware strategies.
> Static schedules (e.g., fixed agent priorities) are fundamentally limited due to the dynamic and context-sensitive nature of LLM interactions. An agent’s utility may vary across different task phases [3, 4], and LLM outputs are inherently stochastic and dependent on evolving context [5, 6], making static heuristics brittle across diverse scenarios.\
> Therefore, robust orchestration requires adaptive mechanisms that dynamically adjust agent priorities based on contextual signals rather than fixed rules. In this light, **we argue that truly effective multi-agent orchestration cannot be static, it must be dynamic and context-aware.**\
> To enable such adaptability, the orchestrator must rely on rich contextual cues, such as intermediate outputs, task progress, and interaction history, which reflect the current state of reasoning [7, 8, 9]. These signals are essential for deciding when to continue, halt, or re-route agent actions. To leverage them effectively, we adopt reinforcement learning, which enables the orchestrator to interpret and act upon these nuanced states (lines 91–92). As a result, our orchestrator dynamically adjusts its strategy based on task requirements, reward signals, and the evolving reasoning states—capabilities fundamentally beyond the reach of static heuristics (lines 91-95, lines 106–117).\
> And we evaluate our method against strong non-RL baselines, including AFlow (MCTS-based) and MacNet (static topologies). As shown in Table 1, our RL-based orchestrator consistently outperforms them across benchmarks, and our work empirically validates RL as an effective approach for orchestrating LLM-powered multi-agent collaboration.\
> **References**\
> [1] Scaling LLM Test-Time Compute Optimally can be More Effective than Scaling Model Parameters. arXiv preprint, 2024.\
> [2] Scaling Large Language Model-based Multi-Agent Collaboration. ICLR, 2025.\
> [3] Why Do Multi-agent LLM Systems Fail? ICLR Workshop, 2025.\
> [4] Generative Agents: Interactive Simulacra of Human Behavior. UIST, 2023.\
> [5] ProSA: Assessing and Understanding the Prompt Sensitivity of LLMs. EMNLP, 2024.\
> [6] Large Language Models Are Latent Variable Models: Explaining and Finding Good Demonstrations for In-Context Learning. NeurIPS, 2023.\
> [7] Language Agents with Reinforcement Learning for Strategic Play in the Werewolf Game. ICML, 2025.\
> [8] Teaching Embodied Reinforcement Learning Agents: Informativeness and Diversity of Language Use. EMNLP, 2024.\
> [9] Efficient Reinforcement Learning with Large Language Model Priors. arXiv preprint, 2024.
> ## To weakness 3:
> **Clarification of Our Orchestration Policy Design**\
> Our orchestration policy uses a pretrained language model under the Bradley-Terry framework—nvidia/Llama-3.1-Nemotron-70B-Reward-HF (see footnote of line 197)—paired with a lightweight three-layer MLP for agent selection. The model acts solely as a semantic encoder: we extract the final hidden state of the last token (an 8192-dimensional embedding) and feed it into an MLP with hidden sizes 512, 128, and 32 to output a score distribution over agents (see lines 997–1007). This setup leverages the model’s strong semantic representations while maintaining efficient, modular decision logic.\
> **Why This Model and Architecture?**\
> We choose Llama-3.1-Nemotron-70B-Reward-HF because it is trained specifically for quality assessment in multi-turn conversations, making it highly effective at encoding fine-grained reasoning signals, and can effectively distinguish between high-quality and low-quality reasoning in generated text [1]. It also achieves state-of-the-art performance on multiple automatic alignment benchmarks (e.g., AlpacaEval 2 LC, Arena Hard, MT-Bench) as of October 2024, surpassing models such as GPT-4o and Claude 3.5 Sonnet [2], thus making it suitable to be our policy's language backbone.\
> We use the final hidden state of the last token—a well-established practice in decoder-only architectures like LLaMA/GPT—as a compact semantic summary of the input. This embedding is projected by a simple three-layer MLP (512 → 128 → 32), enabling dimensionality reduction and feature abstraction, and culminating in a distribution over candidate agents. Similar projection heads are widely used to transform high-dimensional contextual inputs into task-relevant decisions [3-6]. And empirically, this policy network consistently outperforms all baselines across benchmarks (Table 1), validating both our model choice and architecture design.\
> **References**\
> [1] HelpSteer2-Preference: Complementing Ratings with Preferences. ICLR, 2025.\
> [2] NVIDIA. Llama-3.1-Nemotron-70B-Reward-HF. Hugging Face, 2024. \
> [3] iVideoGPT: Interactive VideoGPTs are Scalable World Models. NeurIPS, 2024.\
> [4] InstructEval: Instruction-Tuned Text Evaluator from Human Preference. ACL, 2024.\
> [5] Defending against Indirect Prompt Injection by Instruction Detection. arXiv:2505.06311, 2025.\
> [6] Hugging Face Transformers. LlamaForSequenceClassification Documentation.
> ## To question 2:
> We agree that testing generalizability beyond text-only tasks is important. To this end, we evaluate our method on both code-oriented and embodied interaction tasks. Specifically, we include SRDD [1], which requires generating implementation code from software specifications (see Table 1), and ALFWorld [2], which involves multi-step planning and state tracking in simulated environments (see Section C.4 in appendix). Our method performs well on both, demonstrating strong generalization from language to structured, action-driven domains.\
> **References**\
> [1] ChatDev: Communicative Agents for Software Development. ACL, 2024.\
> [2] ALFWorld: Aligning Text and Embodied Environments for Interactive Learning. ICLR, 2021.
> ## To question 3:
> **For cost-weight λ:**\
> The cost-weight λ controls the trade-off between accuracy and efficiency. A larger λ emphasizes reducing computational cost, potentially at the expense of task performance, whereas a smaller λ prioritizes accuracy but may incur higher computational cost. In extreme settings, an overly large λ can lead to premature termination or under-reasoning, while an excessively small λ may cause unnecessary computation and neglect efficiency. \
> Our sensitivity analysis (in Figure 3) shows (1) λ effectively controls this trade-off, (2) our method is robust within a reasonable range of  λ (e.g., 0.03 and 0.1).\
> Based on these findings, we did not perform exhaustive tuning for λ in each task or setting. Instead, we adopted empirically reasonable values (see Table 3 and Appendix B.3) that ensure balanced performance. While λ was not optimized per task, we believe further task-specific tuning or broader hyperparameter search could enhance the orchestrator’s performance.\
> **For F and φ:**
> - F represents the actual computational cost at each step, measured by real usage metrics such as FLOPs or token count. It is not a tunable parameter but is instead computed from the agent’s execution trace.
> > Typo Correction: We mistakenly referred to F as a “scale factor” in the manuscript — this was a typo. F denotes actual computational cost (e.g., FLOPs or token count). We will correct it in the revised version.
> - φ appears inside a logarithmic term and acts as the denominator of the step count. Typically, φ is set to the maximum episode length. This allows the logarithmic penalty to grow progressively, effectively penalizing prolonged trajectories: at the final step, it reaches log(2), while at step 1, it is near log(1 + 1/φ), which is negligible. Figure 7 analyzes φ’s impact on total cost and success rate. In our experiments, φ is chosen via empirical heuristics and a lightweight grid search (see Table 3 in Section B.3).
> ## To limitation:
> Thanks for the suggestion — we will include all the mentioned points in the revised version, including computation cost and potential limitations in multi-agent collaboration such as mis-coordination and deceptive agreement.

---

> > ### Comment · Reviewer_T1e8 · 2025-08-05
> > **Official Comment by Reviewer T1e8**
> >
> > Thank you for the detailed rebuttal and the additional experiments. These updates substantially strengthen the paper. I will maintain my score. Please ensure the new results and clarifications are incorporated into the final version.

---

> > > ### Author Response · Authors · 2025-08-06
> > >
> > > Thank you very much for your positive feedback and for recognizing the value of our updates. We’re glad that the additional clarifications helped strengthen the paper. We will make sure all new results and explanations are properly incorporated into the final version. Thank you again for your thoughtful review and support!

---

### Official Review · Reviewer_dA6i · 2025-07-07

**Clarity:** 4
**Significance:** 3
**Originality:** 3
**Rating:** 5
**Confidence:** 4

**Summary:**

* Aims to learn a dynamic orchestrator for LLM-based multi-agent collaboration, using a "puppeteer-style" paradigm

**Questions:**

* I'd like to know how this work is related to works in MARL coordination; e.g. ALMA [1], which is a similarly hierarchical approach which also performs "orchestration" (task allocation) dynamically; it seems that there's probably some overlap of ideas (if not with ALMA, then more broadly in that literature)

[1] https://arxiv.org/abs/2205.14205

**Ethical Concerns:**

["NO or VERY MINOR ethics concerns only"]

**Final Justification:**

I stand by my initial review and therefore maintain my current score.

**Limitations:**

yes, Appendix E; appears comprehensive, and encapsulates my thoughts

**Quality:**

4

**Strengths And Weaknesses:**

Strengths
* I like the idea of evolving an orchestrator; I agree that many paradigms for coordination are too rigid
* The paper is well-written; clear and easy to follow; the illustrations are very useful as well
* The evaluation tasks seem reasonable, and the performance improvements are clear
* Extensive interpretable analysis of experiments

Weaknesses
* I can't help but feel that there must be much more literature on this kind of orchestration that is not referenced in the paper (see my question below); especially pre-LLM multi-agent RL (MARL) literature

---

> ### Author Rebuttal · Authors · 2025-07-30
>
> **Thank you for your insightful comment and kind recognition.**\
> Our paper mainly includes related works that leverage reinforcement learning to optimize and orchestrate LLM-based agents. While our current work primarily focuses on LLM-based multi-agent systems, and thus does not emphasize pre-LLM MARL literature extensively, we fully acknowledge that many of the coordination principles and theoretical foundations established in the MARL community continue to inspire the design of LLM-based agents, such as ALMA[1], RODE[2], and other[3,4]. In the revised version, **we will incorporate relevant citations and explicitly highlight both the inspirations and distinctions between classical MARL and emerging LLM-based paradigms.**
>
> **References**\
> [1] ALMA: Hierarchical Learning for Composite Multi-Agent Tasks. Conference on Neural Information Processing Systems (NeurIPS), 2022.\
> [2] RODE: Learning Roles to Decompose Multi-Agent Tasks. International Conference on Learning Representations (ICLR), 2021.\
> [3] Coach-Player Multi-Agent Reinforcement Learning for Dynamic Team Composition. International Conference on Machine Learning (ICML), 2021.\
> [4] Options as Responses: Grounding Behavioural Hierarchies in Multi-Agent Reinforcement Learning. International Conference on Machine Learning (ICML), 2020.

---

### Note · Authors · 2025-08-12

Dear Area Chair and Reviewers,

We are grateful for the thorough reviews and constructive discussion, which have been important in enhancing the paper's clarity and rigor.

Our core contribution is a novel framework for LLM-based multi-agent collaboration that overcomes the limitations of static approaches through two key innovations. First, **Dynamic Orchestration** employs a centralized and context-aware orchestrator to flexibly activate agents ***(acknowledged by all reviewers)***. Second, **Adaptive Evolution** uses reinforcement learning to continuously refine the orchestration policy based on task feedback, progressively pruning inefficient paths and highlighting effective and efficient ways ***(acknowledged by Reviewer dA6i, Reviewer T1e8, Reviewer QYtQ)***. This dual approach is empirically validated across diverse benchmarks, consistently achieving more effective solutions with significantly less computational overhead. Crucially, our analysis reveals these gains stem from the emergent formation of compact and efficient cyclic reasoning structures, offering a new insight into effective agent collaboration.

Finally, we wish to address the critical point raised by Reviewer XZCy regarding the necessity of RL over a simpler LLM-based orchestrator. This is indeed a baseline, representing a 'zero-shot' generalist approach. However, the performance of such a static orchestrator is fundamentally capped by its pre-trained knowledge. It may perform well, but it cannot learn from its successes and failures on a specific task to become better. Our work's core contribution lies precisely in breaking through this ceiling. The essence of our framework is not merely being 'dynamic', but being 'optimizable' and 'adaptive'.
Our RL-based approach enables the orchestrator to evolve from a general-purpose model into a domain-specific expert. By learning from task rewards, it discovers non-obvious, more efficient collaborative strategies that a static LLM, relying only on its generalized world knowledge, could never find. This evolutionary capability is the fundamental justification for RL's necessity, as it creates a system that truly learns and surpasses its initial limitations.

**We are encouraged by the broad recognition of our paper's novelty and impact. We are fully committed to incorporating all feedback for the revised version. To support reproducible research, our code will be made public upon acceptance. Thank you again for your time and consideration.**

---

### Decision · Program_Chairs · 2025-09-17

**Decision:**

Accept (poster)

**Comment:**

In this submission, the authors propose a puppeteer-style paradigm for LLM-based multi-agent collaboration whereby a centralized orchestrator dynamically directs agents as a response to task states that keep evolving over time. Trained using RL to adaptively order and prioritize the agent, the orchestrator provides flexible and evolvable collective reasoning. The authors provide empirical results on closed- and open-domain scenarios which show that the proposed method achieves superior performance with reduced computational costs.

The reviewers have noted several strengths of the paper, including impressive, cost-aware improvements over four diverse benchmarks. The reviewers agree that the paper provides a clear and compelling motivation behind the main idea, and that framing the multi-agent collaboration problem as a centralized & learnable sequential decision process is conceptually clean. During the rebuttal period the authors addressed many of the concerns raised by the reviewers. I believe that this is an interesting research direction with compelling results, and therefore I’m pleased to recommend it for acceptance.